# WHEN DOES COMPOSITIONAL STRUCTURE YIELD COMPOSITIONAL GENERALIZATION? A KERNEL THEORY.

**Samuel Lippl**
Center for Theoretical Neuroscience
Columbia University
New York, NY, USA
samuel.lippl@columbia.edu

**Kimberly Stachenfeld**
Google DeepMind and
Center for Theoretical Neuroscience
Columbia University
New York, NY, USA
stachenfeld@deepmind.com

## ABSTRACT

Compositional generalization (the ability to respond correctly to novel combinations of familiar components) is thought to be a cornerstone of intelligent behavior. Compositionally structured (e.g. disentangled) representations support this ability; however, the conditions under which they are sufficient for the emergence of compositional generalization remain unclear. To address this gap, we present a theory of compositional generalization in kernel models with fixed, compositionally structured representations. This provides a tractable framework for characterizing the impact of training data statistics on generalization. We find that these models are limited to functions that assign values to each combination of components seen during training, and then sum up these values ("conjunction-wise additivity"). This imposes fundamental restrictions on the set of tasks compositionally structured kernel models can learn, in particular preventing them from transitively generalizing equivalence relations. Even for compositional tasks that they can learn in principle, we identify novel failure modes in compositional generalization (memorization leak and shortcut bias) that arise from biases in the training data. Finally, we empirically validate our theory, showing that it captures the behavior of deep neural networks (convolutional networks, residual networks, and Vision Transformers) trained on a set of compositional tasks with similarly structured data. Ultimately, this work examines how statistical structure in the training data can affect compositional generalization, with implications for how to identify and remedy failure modes in deep learning models.

## 1 INTRODUCTION

Humans' understanding of the world is inherently compositional: once familiar with the concepts "pink" and "elephant," we can immediately imagine a pink elephant. Stitching together concepts in this way lets humans generalize far beyond our prior experience, allowing us to cope with unfamiliar situations and imagine things that do not yet exist (Fig. 1a) (Lake et al., 2017; Frankland & Greene, 2020). Understanding the basis of compositional generalization in humans and animals, and building it into machine learning models, is a long-standing and historically vexing problem (Fodor & Pylyshyn, 1988; Battaglia et al., 2018; Lake & Baroni, 2018; Lepori et al., 2023). While a wide range of studies have investigated the conditions under which compositionally structured (i.e. "disentangled") representations can be learned (Hinton et al., 2011; Higgins et al., 2017; Träuble et al., 2021; Whittington et al., 2023; Ren et al., 2023), it remains unclear when learning these representations is actually useful to a downstream neural network. Some work suggests that disentangled representations improve compositional generalization (Esmaeili et al., 2019; van Steenkiste et al., 2019). Other studies challenge this view (Locatello et al., 2019; Schott et al., 2022). In general, a systematic understanding of the relationships among compositional tasks remains elusive (Hupkes et al., 2020).

To make progress on this question, we focus on standard statistical learning, a fundamental basis for generalization that affects almost any machine learning model. Generalization in statistical learning depends on the similarities between different inputs. In compositionally structured representations, inputs that have components in common (say, a red circle and a blue circle) are more similar to each

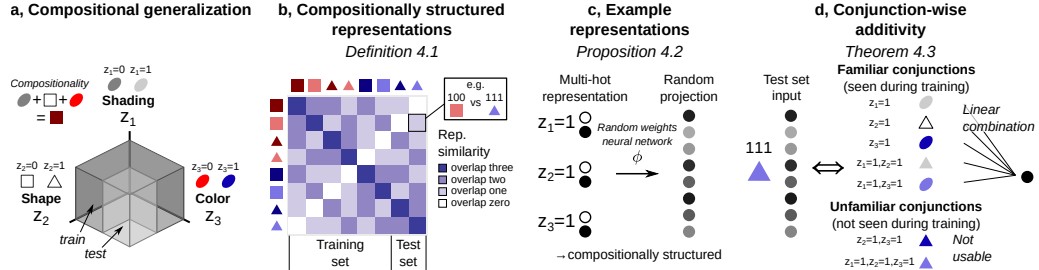

Figure 1: Overview of main theoretical findings (Section 4). **a**, We consider inputs with several categorical components and study compositional generalization to novel combinations of components. **b**, We assume compositionally structured representations for which trials with the same number of overlaps have identical similarities. **c**, We find that a random weights neural network conserves compositional structure: if its input is compositionally structured then so is its output. **d**, We find that compositionally structured kernel models are constrained to adding up values for each conjunction seen during training ("conjunction-wise additivity").

other than inputs that do not (say, a red circle and a blue square). As a result, the compositional structure of a task is reflected in its dataset statistics, influencing the model's generalization behavior.

To understand when and how statistical learning leads to successful compositional behavior, we developed a theory for the behavior of kernel models on compositional tasks. Kernel models are an important class of statistical models that are able to provide a simplified approximation to neural networks while maintaining analytical tractability (Jäkel et al., 2008; 2009). For example, kernel models accurately describe the behavior of learning and fine-tuning in neural networks under certain conditions (Jacot et al., 2018; Malladi et al., 2023). More broadly, they provide a tractable framework for characterizing the impact of dataset statistics on generalization (Canatar et al., 2021a;b).

Despite their broad relevance and relative simplicity, it has remained unclear how even kernel models generalize on compositional tasks (though see Abbe et al., 2023; Lippl et al., 2024, see Section 2). To address this gap, we present a theory of compositional generalization in kernel models. We define a general class of "compositionally structured representations" (Fig. 1b) and a general family of compositional tasks with no constraints on the input-output mapping (ensuring broad applicability of our theory). We then theoretically characterize the compositional behavior of compositionally structured kernel models, and go beyond kernel models to empirically validate our theory in several relevant deep neural network architectures. Our specific contributions are as follows:

- In Section 4, we show that compositionally structured kernel models are constrained to summing up values implicitly assigned to each component or combination of components seen during training (Fig. 1). We call such computations "conjunction-wise additive."

- In Section 5, we then characterize how representational geometry determines whether kernel models will generalize successfully on conjunction-wise additive compositional tasks, highlighting two important failure modes (memorization leak and shortcut bias). This demonstrates that disentangled representations are not sufficient for downstream compositional generalization (even on conjunction-wise additive tasks) and explains why.

- Finally, in Section 6, we validate our theory in several deep neural network architectures, showing that it captures their behavior on conjunction-wise additive tasks.

Overall, we take a step towards a general theory of compositional generalization. Our theory systematically clarifies the compositional generalization behavior of deep networks and lays a foundation for the design of new learning mechanisms that overcome their limitations.

## 2 RELATED WORK

**Compositional generalization.** Compositionality is an important theme in both human and machine cognition, including visual reasoning (Lake et al., 2015; Johnson et al., 2017; Schwartenbeck et al., 2023), language production (Hupkes et al., 2020), and rule learning (Ito et al., 2022; Abdool et al.,

2023). While recent breakthroughs have led to massive improvements in models' compositional capacities, these models can still fail spectacularly (Srivastava et al., 2023; Lewis et al., 2023; West, 2023). Attempts to improve compositional generalization often leverage meta-learning (Mitchell et al., 2021; Wu et al., 2023; Lake & Baroni, 2023) or modular architectures (Andreas et al., 2017). Constraining a network's compositional function (e.g. to adding up a value for each component) can guarantee modular specialization and compositional generalization (Lachapelle et al., 2023; Wiedemer et al., 2023a;b; Schug et al., 2024). However, these networks are extremely limited in the tasks they can learn (as we will show below) and end-to-end training of modular architectures without such constraints often does not result in the desired specialization (Bahdanau et al., 2019; Mittal et al., 2022; Jarvis et al., 2023).

**Kernel regime.** When using gradient descent, ridge regression, or similar learning algorithms to train the readout weights of a model $f_w(x) = w^T\phi(x)$, the behavior of that model can be described in terms of the kernel $K_\phi(x, x') = \phi(x)^T\phi(x')$ induced by $\phi(x)$ (Schölkopf, 2000). Notably, when trained on a dataset $\{(x_i, y_i)\}_{i=1}^n$, such "kernel models" can be written in dual form as $f_a(x) = \sum_{i=1}^n a_i K_\phi(x, x_i)$, for inferred dual coefficients $a \in \mathbb{R}^n$. Prior work has shown that neural networks with large initial weights or wide architectures are well approximated by a kernel model ("kernel regime") (Jacot et al., 2018). In contrast, the "feature-learning regime" (which can be brought forth, for example, by small initial weights or small width) yields more substantial changes in neural networks' internal representations (Chizat et al., 2019).

**Norm minimization.** Gradient descent, ridge regression, and neural networks in the kernel regime learn the readout weights that describe the training data with minimal $\ell_2$-norm (Soudry et al., 2018; Gunasekar et al., 2018; Ji et al., 2020). Norm minimization is a standard theoretical framework for analyzing how representational geometry and dataset statistics influence generalization (Canatar et al., 2021a;b; 2023) and we here apply it to the compositional task setting. This is similar to Lippl et al. (2024) who characterize model behavior on a specific compositional task (transitive ordering) and Abbe et al. (2023) who characterize the inductive bias of norm minimization for inputs with binary components in the limit of infinite components. Compared to these prior works, we analyze a broader range of compositional tasks and derive exact constraints for finite numbers of components.

**Memorization leak.** Machine learning models sometimes memorize (parts of) their training data instead of learning a generalizable rule (Zhang et al., 2020; Dasgupta et al., 2022). This may improve generalization on long-tailed data (Feldman, 2020), but often impairs out-of-distribution performance (Elangovan et al., 2021). We find that models partially memorize their training data even when they extract the correct rule. Jarvis et al. (2023) analyze a related phenomenon in deep linear networks.

**Shortcut learning.** Shortcut learning refers to models exploiting spurious correlations between certain features of the input data and the target (Shah et al., 2020; Nagarajan et al., 2021). This substantially impacts their performance out of distribution, where those correlations may not hold (Geirhos et al., 2020). We theoretically analyze how shortcut biases impact compositional generalization.

## 3 MODEL AND TASK SETUP

### 3.1 TASK SPACE

We consider an input $x$ representing a set of underlying components $z = (z_c)_{c=1}^C$, where each component $z_c \in Z_c$ is drawn from a discrete, finite set of possible components. For example, $x$ could be a "multi-hot" concatenation of one-hot representations of all components (Fig. 1c). We make a specific assumption about how the trial-by-trial similarity of input representations $x$ relates to the underlying components $z$ (Fig. 1b):

**Definition 3.1.** A representation $x$ is "compositionally structured" iff its kernel $K(x, x') = x^T x'$ only depends on the number of components that are identical between $z$ and $z'$, where $z$ and $z'$ are the components represented by $x$ and $x'$ respectively.

In particular, the multi-hot representation described above is compositionally structured, but we will see below that this concept captures a richer set of representations as well. The target, $y \in \mathbb{R}$, is given by an arbitrary function of $z$, ensuring that our framework is agnostic to the underlying compositional structure. After training models on certain combinations of components $Z^{\text{train}} \subset Z = \prod_{c=1}^C Z_c$, we assess generalization on all other combinations $Z^{\text{test}} := Z \setminus Z^{\text{train}}$.

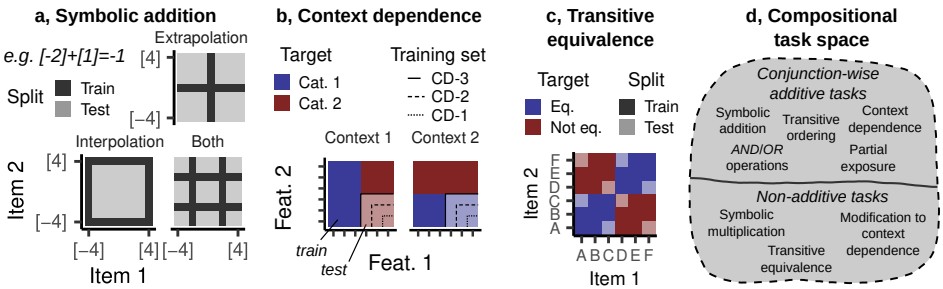

Figure 2: The compositional task space. **a**, Example training sets for symbolic addition. The grid represents pairs of components with associated values $-4, \ldots, 4$. The training set consists of certain rows and columns of this grid. **b**, Context dependence. In context 1, feat. 1 determines the category; in context 2, feat. 2 determines the category. The training sets leave out subsets of the lower right orthant. **c**, Transitive equivalence: six items are split up into two equivalence classes (e.g. A,B,C and D,E,F) and generalization requires transitive inference over equivalence classes (e.g. $A = B$ and $B = C$ implies $A = C$). **d**, Our theory partitions the task space into conjunction-wise additive (which can be solved by kernel models) and non-additive tasks (which cannot).

Our theory characterizes constraints on model behavior for arbitrary tasks within this family, regardless of the ground-truth input-output mapping. Below we describe three example tasks that represent important building blocks for compositional reasoning in machine learning and cognitive science. We will focus on these tasks in the main text, to illustrate our general theory and analyze its consequences in more detail. In Appendix E, we describe additional tasks captured by our theory (Fig. 2d).

**Symbolic addition.** Many tasks involve inferring a magnitude associated with different underlying components (e.g. adding handwritten digits) (Lorenzi et al., 2021; Sheahan et al., 2021). We consider two components $(z_1, z_2)$ with hidden values $v_1(z_1)$ and $v_2(z_2)$. The target is the sum of those values: $y = v_1(z_1) + v_2(z_2)$. After sufficient exposure to individual items in different combinations, an additive model could in principle generalize to novel combinations. We consider training sets consisting of entire rows and columns of the compositional dataset (Fig. 2a), varying the number of rows and columns as well as whether generalization involves interpolation or also extrapolation.

**Context dependence.** The relevance of different stimuli often depends crucially on our current context (Bouton et al., 1999; Taylor & Ivry, 2013; Parker & Hollister, 2014; Ito et al., 2022). We consider a task with three input components $(z_{co}, z_{f1}, z_{f2})$. The context $(z_{co})$ has two possible values specifying whether $z_{f1}$ or $z_{f2}$ determine the response. Features have six possible values which are divided into two classes. If the model has learned this context dependence, it should generalize to novel feature combinations. We evaluate on the subset of data for which $z_{f1}$ indicates Cat. 2 and $z_{f2}$ indicates Cat. 1. In the most extreme generalization test (*CD-3*), we leave out the entire subset; in easier versions, we leave out combinations of one or two features (*CD-1*, *CD-2*) (Fig. 2b).

**Transitive equivalence.** Relational reasoning involves extending learned relations to new item combinations (Halford et al., 2010; Battaglia et al., 2018). Given an unobserved (and arbitrary) equivalence relation, the task here is to determine whether two presented items $(z_1, z_2)$ are equivalent. The model should generalize to novel item pairs using transitivity ($A = B$ and $B = C$ imply $A = C$) (Fig. 2c). This is an important instance of relational cognition (often studied as "associative inference" in cognitive science (Schlichting & Preston, 2015; Spalding et al., 2018)). Although prior work has found that kernel models often successfully generalize on a transitive *ordered* relation ($A > B$ and $B > C$ imply $A > C$, Lippl et al., 2024), the behavior of kernel models on equivalence relations has remained unclear.

## 3.2 MODELS

Our theory characterizes kernel models with a compositionally structured input $x \in \mathbb{R}^d$ that apply a transform $\phi(x) \in \mathbb{R}^h$ and learn a linear readout, $f_w(x) := w^T \phi(x)$, using gradient descent with initial weights $w_0 = 0$. For $\phi$, we consider a neural network with random weights (in the infinite-width limit; see Appendix A.3). Our setup captures, for example, the random feature model studied by Abbe et al. (2023) and training via backpropagation in the kernel regime.

## 4 KERNEL MODELS ARE CONJUNCTION-WISE ADDITIVE

Our primary theoretical contribution is to characterize the full range of compositional computations that can be implemented by kernel models with compositionally structured inputs. We find that even though these models can learn arbitrary training sets, their test set behavior is restricted to adding up values for each conjunction (combination of components) seen during training. We call this motif "conjunction-wise additivity." Below, we formally state our finding and explain its implications.

### 4.1 RANDOM NETWORKS YIELD COMPOSITIONALLY STRUCTURED REPRESENTATIONS

We first note that a linear readout of the multi-hot input constrains the model to adding up a value for each component ("component-wise additivity"): $f(x) = \sum_{c=1}^{C} f_c(z_c)$. Although models of this class perfectly generalize on component-wise additive tasks, such as symbolic addition, they are incapable of even learning the *training data* (much less generalizing) on tasks that are not component-wise additive. In particular, these models cannot learn context-dependent computations or equivalence relations — both fundamental instances of compositional reasoning.

The nonlinear transform $\phi(x)$ can overcome this constraint; in particular, multi-layer nonlinear neural networks can learn arbitrary training data (in the infinite-width limit) (Hornik et al., 1989; Cybenko, 1989; Rigotti et al., 2013). Normally, the wide range of possible network architectures makes it difficult to derive general statements about their representations, but in this case we can take advantage of the fact that $\phi(x)$ will also be compositionally structured (Fig. 1c):

**Proposition 4.1.** *For a random weights neural network $\phi$ with a compositionally structured input $x$, in the infinite-width limit, $\phi(x)$ is also compositionally structured.*

Proposition 4.1 holds because in the infinite-width limit, the kernel of random weight neural networks depends only on the input kernel (Cho & Saul, 2009, see Appendix A.3). We now consider the constraints that compositional structure imposes on the models' generalization behavior.

### 4.2 COMPOSITIONALLY STRUCTURED KERNEL MODELS ARE CONJUNCTION-WISE ADDITIVE

We find that any kernel model with a compositionally structured representation is constrained to be conjunction-wise additive. To formally state this finding, we define, for each $z \in Z$, the set of conjunctions for which $z$ overlaps with some element in the training set $z^{\text{tr}} \in Z^{\text{train}}$,

$$\text{Conj}(z|Z^{\text{train}}) := \left\{ J \subseteq \{1, \dots, C\} | \text{there is } z^{\text{tr}} \in Z^{\text{train}} \text{ such that } z_c = z_c^{\text{tr}} \text{ for all } c \in J \right\}. \quad (1)$$

**Theorem 4.2.** *For any kernel model $f$ with a compositionally structured representation, we can find conjunction-wise functions $f_J : \prod_{c \in J} Z_c \to \mathbb{R}$, where $J \subseteq \{1, \dots, C\}$, such that for any input $x \in \mathbb{R}^d$ representing components $z \in Z$, the model response is given by*

$$f(x) = \sum_{J \in \text{Conj}(z|Z^{\text{train}})} f_J(z_J), \quad z_J := (z_c)_{c \in J}. \quad (2)$$

We prove the theorem in Appendix A.4. It implies that for a given test input, kernel models add up a value for each component and partial conjunction seen during training (Fig. 1d). We call this computation "conjunction-wise additive." In particular, for inputs with two components, the model's behavior on the training set can be expressed as $f(x) = f_1(z_1) + f_2(z_2) + f_{12}(z_1, z_2)$. Thus, $f_{12}(z_1, z_2)$ enables the model to learn arbitrary training data. However, if $x$ represents $z \in Z^{\text{test}}$, the training set does not contain any input with the conjunction $(z_1, z_2)$ and the model is component-wise additive: $f(x) = f_1(z_1) + f_2(z_2)$. For inputs with more than two components, a conjunction-wise additive computation additionally encodes partial conjunctions seen during training, e.g., if $Z^{\text{train}}$ contains the conjunction $(z_1, z_2)$, $f(x) = f_1(z_1) + f_2(z_2) + f_3(z_3) + f_{12}(z_1, z_2)$.

### 4.3 CONJUNCTION-WISE ADDITIVITY CONSTRAINS THE TASKS KERNEL MODELS CAN SOLVE

Theorem 4.2 implies that compositionally structured kernel models can only solve tasks that can be expressed in conjunction-wise additive terms. This highlights a fundamental computational restriction, partitioning the compositional task space into tasks that can and cannot be solved by kernel models

(Fig. 2d). Intriguingly, these restrictions are not caused by an architectural constraint, as the model can learn arbitrary training data (at least with a nonlinear transformation $\phi$, see Section 5.1). Rather, we highlight restrictions on how models – without any architectural constraints – can *generalize* (Zhang et al., 2021). We now spell out the consequences of the highlighted restrictions.

First, for inputs with two components, we noted above that the model's behavior on the test set is component-wise additive. A consequence of this is that compositionally structured models can only generalize on component-wise additive tasks. In particular, this implies that these models cannot generalize on transitive equivalence — a fundamental instance of relational reasoning. Intriguingly, transitive ordering – a superficially similar task – can be solved by a kernel model. This highlights the importance of a formal perspective on compositional tasks.

More generally, to see whether a kernel model can, in principle, generalize on a task with more than two input components, we must 1) identify, for each $z \in Z^{\text{test}}$, the conjunctions seen during training, $\text{Conj}(z|Z^{\text{train}})$, and 2) determine whether the target can be written as a conjunction-wise sum (Eq. (2)). For context dependence for example, for all $z \in Z^{\text{test}}$, we have seen the context-feature conjunctions 12 and 13, but not the feature-feature conjunction 23. As a result, model behavior on a test trial $x$ representing the components $z_{\text{co}}, z_{f1}, z_{f2}$ can be written as $f(x) = f_1(z_{\text{co}}) + f_2(z_{f1}) + f_3(z_{f2}) + f_{12}(z_{\text{co}}, z_{f1}) + f_{13}(z_{\text{co}}, z_{f2})$. Importantly, this conjunction-wise additive function can encode context dependence, specifically with the following schema: $f_{12}(z_{\text{co}}, z_{f1})$ encodes the target when the context $z_{\text{co}}$ indicates $z_{f1}$ as relevant and is zero otherwise; $f_{13}(z_{\text{co}}, z_{f2})$ works in the opposite way. Thus, the model can, in principle, identify a set of weights that would allow it to generalize correctly.

This example illustrates that conjunction-wise additivity tells us both *whether* kernel models with compositionally structured representations can solve a certain task, and also *how* they solve it. It even tells us how to make a task non-additive: in Appendix E.2.3, we describe such a modification to context dependence that tests generalization to novel context-feature conjunctions. By helping us design hard benchmark tasks that are not solvable by kernel models, our framework grounds research into learning mechanisms that can implement other compositional computations.

What do our findings imply about compositional generalization in more general representations? In this section, we analyzed compositionally structured inputs, as we consider them the best-case scenario for compositional generalization. Accordingly, we expect that the limitations revealed by our theory are also relevant for kernel models with non-compositionally structured representations. We present two pieces of preliminary evidence for this conjecture in the appendix: First, we theoretically prove that randomly sampled representations (that are only compositionally structured in expectation) are also conjunction-wise additive in expectation (Appendix A.5). Second, we empirically investigate compositional generalization in disentangled representation learning models trained on the DSprites dataset (Locatello et al., 2019), finding that, when averaged across randomly sampled task instances, these models are also limited to conjunction-wise additive computations (Appendix C). These findings suggest that the generalization class we have highlighted in this section captures fundamental limitations of kernel models beyond our theoretical setting.

## 5 REPRESENTATIONAL GEOMETRY STILL IMPACTS GENERALIZATION ON CONJUNCTION-WISE ADDITIVE TASKS IN KERNEL MODELS

Conjunction-wise additivity only determines whether kernel models can, in principle, solve a certain task. However, model generalization also depends on whether the model identifies the correct conjunction-wise function. This depends on the model's representational geometry and the training data statistics. As noted in Section 2, the behavior of kernel models depends on their representation $\phi(x)$ only through the induced kernel $K_\phi(x, x')$. In this section, we first investigate how different choices in network architecture influence its induced kernel (Section 5.1). Then, we characterize how this kernel impacts task performance on symbolic addition and context dependence (Section 5.2).

### 5.1 OVERLAP SALIENCE CHARACTERIZES COMPOSITIONAL REPRESENTATIONAL GEOMETRY

To characterize compositionally structured representations, we introduce a new metric: representational salience. We formally define this metric in Appendix B.1. Intuitively, for $k = 1, \ldots, C$, the salience $S(k; C)$ measures the unique contribution of the subpopulation representing a conjunction of $c$ components. Further, $S(k; C)$ is normalized so that all saliences sum up to one. For example,

the multi-hot input only encodes single components and therefore $S(1; C) = 1/C$ and, for all $k > 1$, $S(k; C) = 0$. Note that the model can only use conjunctions whose salience is nonzero. For example, the multi-hot input is constrained to a component-wise sum (as all other saliences are zero).

$S(k; C)$ is computed from the representational similarities $K_\phi(x, x') = \phi(x)^T \phi(x')$. This metric is useful because unlike $K_\phi(x, x')$, it makes immediately apparent how strongly different conjunctions are encoded (Appendix B.2). Furthermore, it reduces the $C + 1$ similarities characterizing the space of compositionally structured representations to $C - 1$ free parameters, removing the overall magnitude of the representation (by normalizing) and the baseline activity (by not considering the similarity between inputs with no overlap). This is because these parameters often have negligible impact on model behavior (Appendix B.3).

We now analyze how $S(k; C)$ evolves over different layers of a random network, using three components as an example (Fig. 3; see Appendix B.4 for other values of $C$). The input only represents individual components and so $S(2; 3)$ and $S(3; 3)$ are zero. As the network gets deeper, these saliences increase and $S(1; 3)$ decreases. In particular, because $S(3; 3) > 0$ for all networks with at least one layer, the resulting representations can learn arbitrary training data (using the full conjunction). As depth increases, $S(2; 3)$ eventually decreases again, whereas $S(3; 3)$ continues to increase.

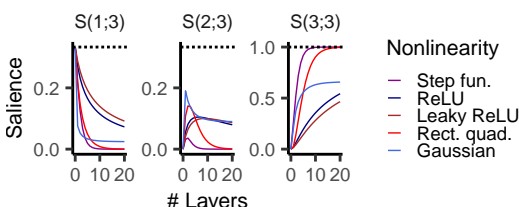

Figure 3: Representational salience (for three input components) in a random weight neural network with variable numbers of layers and nonlinearities.

In fact, we prove that for ReLU networks in the limit of infinite depth, $S(C; C) \to 1$ (Proposition B.2). Put differently, very deep networks exclusively encode the full conjunction. As a result, these models effectively implement a look-up table, memorizing the training data without generalizing to novel combinations of components. In the next section, we find that even a moderately conjunctive representation presents significant challenges to compositional generalization.

## 5.2 KERNEL MODELS SUFFER FROM MEMORIZATION LEAK AND SHORTCUT BIAS

While there are many readout weights that can lead to minimal error, kernel models trained with gradient descent learn the weights with minimal $\ell_2$-norm (see Section 2). We now characterize how this inductive bias influences compositional generalization. To characterize the full range of compositionally structured representations, we compute the behavior of the model directly using the kernel (see Appendix D). Importantly, our analysis clarifies the behavior of all random weight neural networks, including those covered in the previous section.

**Symbolic addition suffers from a memorization leak.** We first consider an example task ($z_1, z_2 \in \{[-4], \ldots, [4]\}$; the training set consists of all trials containing a $[0]$, Fig. 2a) and an example representation ($S(1; 2) = 0.4$). We find that though the model perfectly learns the training cases, it underestimates the test cases by a proportional factor (Fig. 4a). To understand why, we consider the model's functional form on the training cases: $f(x) = f_1(z_1) + f_2(z_2) + f_{12}(z)$ (see Section 4). Intuitively, $\ell_2$-norm minimization tends to yield distributed weights. As a result, unless the salience of the conjunction $S(2; 2)$ is zero, the kernel model will associate some non-zero weight with it (i.e. $f_{12}(z_1, z_2) \neq 0$). This distorts the test set generalization, $f(x) = f_1(z_1) + f_2(z_2)$.

We call the tendency to use the full conjunction (and its detrimental effect on generalization) "memorization leak." To characterize it more systematically (and add mathematical rigor to the intuition above), we analytically characterize model behavior on a general family of symbolic addition tasks:

**Proposition 5.1.** *Consider inputs $z_1, z_2 \in \{[v]\}_{v \in \mathcal{V}}, \mathcal{V} \subset \mathbb{R}$, with associated values $v$. We assume that the training set contains all pairs such that at least one component is $z_c \in \{[w]\}_{w \in \mathcal{W}}, \mathcal{W} \subset \mathcal{V}$ and that the average value in both $\mathcal{V}$ and $\mathcal{W}$ is zero. Then, model behavior on the test set is given by*

$$f([v_1], [v_2]) = m(v_1 + v_2), \quad m := \frac{p \cdot S(1;2)}{1 + (p-2)S(1;2)}, \quad p := |\mathcal{W}| \tag{3}$$

We prove the proposition in Appendix E.1. It implies that for this entire task family, model generalization is distorted by a proportional factor $m$ (Fig. 4b). The formula implies that $m < 1$ as long as

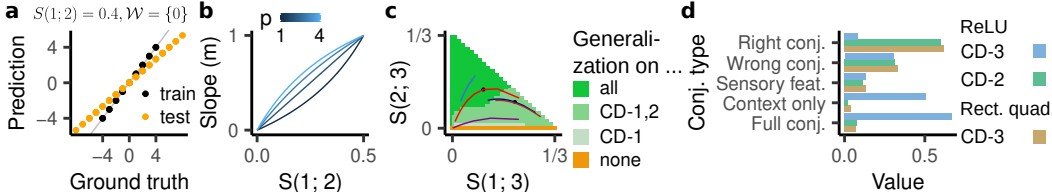

Figure 4: Kernel models' behavior on *(a,b)* symbolic addition and *(c,d)* context dependence. **a**, Model predictions on training and test set plotted against ground truth for an example case ($S(1;2) = 0.4$, $\mathcal{W} = \{0\}$). **b**, The slope of the test set as a function of $S(1;2)$ and the training set size $p = |\mathcal{W}|$. **c**, Generalization on context dependence as a function of representational salience. Trajectories of networks with different nonlinearities are highlighted (color scale see Fig. 3). **d**, Coefficients of the different conjunction types for two example networks with three layers and different nonlinearities.

$S(1;2) < \frac{1}{2}$ (i.e. as long as $S(2;2) > 0$). Further, $m$ is smaller for lower $S(1;2)$ and smaller training set size $p$. (Note that $\mathcal{W}$ corresponds to the rows/columns making up the example training sets in Fig. 2a.) Interestingly, these are the only two factors that influence $m$. In particular, even though interpolation is often seen as easier than extrapolation, this does not impact model behavior here.

More broadly, the memorization leak is a ubiquitous issue for statistical learning models trained on compositional tasks: $\ell_2$-norm minimization tends to yield distributed weights and so if the conjunctive population is represented, the model will generally end up partially relying on this conjunction. As shown above, this necessarily distorts generalization. This issue also arises in tasks that involve directly decoding specific components, a popular task for evaluating disentangled representations (Locatello et al., 2019; Schott et al., 2022) (see Appendix E.3 for a minimal example).

**Context dependence suffers from a shortcut bias.** Next, we empirically analyzed model generalization on context dependence across different representational geometries and training sets. We found that on a given task, each model either generalizes with 100% or 0% accuracy. For *CD-3* (the task version leaving out the largest subset of feature combinations), the model only generalizes when $S(2;3)$ is high relative to $S(1;3)$ (Fig. 4c, dark green area). As a result, compositional generalization is highly sensitive to the nonlinearity and depth of the network. In contrast, for *CD-2* and *CD-1*, a much wider range of representational geometries generalizes successfully.

To understand this phenomenon, we determined the total magnitude of model weights associated with the different conjunctions (Fig. 4d). We found that unsuccessful models (e.g. the blue color in Fig. 4d) had much larger magnitudes associated with the context component and the full conjunction. Notably, on *CD-3*, context is highly correlated with the target and can predict $\frac{2}{3}$ of the training data. Models with high $S(1;3)$ (context) and $S(3;3)$ (full conjunction) exploit this context shortcut and use the full conjunction to learn the remaining training data. This strategy explains why these models fail to generalize to the test set. For *CD-2*, context is much less predictive on the training data (accuracy of $\frac{9}{16}$). This explains why only very low $S(2;3)$ yields failure to generalize on *CD-2* or *CD-1*.

Our analysis shows how model and task structure interact to either give rise to a generalizable rule or a statistical shortcut. This highlights that for compositional tasks with strong spurious correlations, model behavior will be highly sensitive to architectural hyperparameters affecting the representational geometry such as depth and nonlinearity. Indeed, this sensitivity to minor experimental choices could explain why the literature has been so divided on the usefulness of disentangled representations.

## 6 OUR THEORY CAN DESCRIBE THE BEHAVIOR OF DEEP NETWORKS ON CONJUNCTION-WISE ADDITIVE TASKS

So far, our analyses have been limited to simple kernel models, for their analytical tractability. We next tested whether the insights developed from these simple models extend to a broader class of models: large-scale neural networks. We considered symbolic addition and context dependence on inputs created by concatenations of images from MNIST (Lecun et al., 1998) or CIFAR-10 (Krizhevsky et al., 2009) (see Appendix D). Each component corresponds to random samples from a particular category rather than the one-hot input considered so far and we study both in-distribution

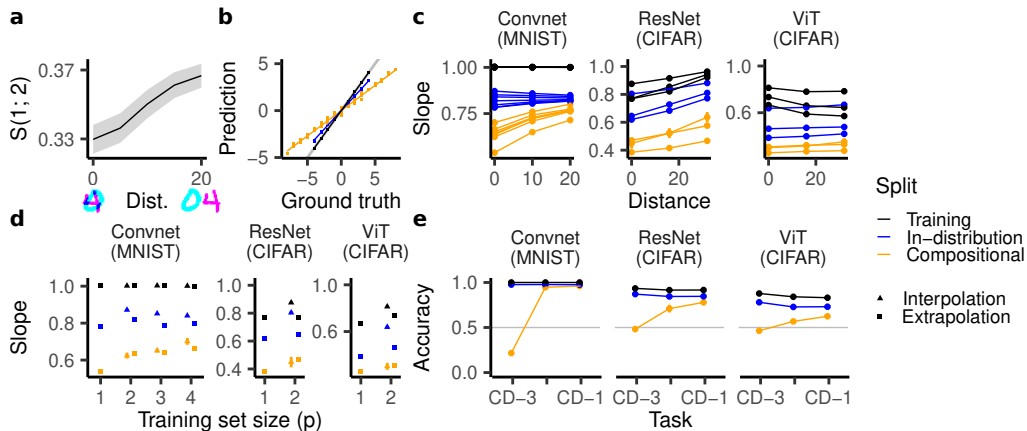

Figure 5: Testing our theory in deep networks trained on MNIST and CIFAR versions of compositional tasks. Ranges indicate mean $\pm$ one std. error (often too small to be visible). **a**, $S(1; 2)$ in an intermediate ConvNet layer for different distances between digits. Lower distance yields a more conjunctive representation. **b**, Average model prediction for each combination of components plotted against the ground truth (MNIST, distance of zero, $\mathcal{W} = \{0\}$). Generalization on the compositional split is distorted by a proportional factor. **c, d**, Slope of this linear relationship across all datasets as a function of $c$, distance (each line corresponding to a particular $\mathcal{W}$) and **d**, training set (for a distance of zero). For MSE instead of slope, see Fig. 12. **e**, Accuracy on all variants of context dependence.

and compositional generalization. Notably, different image categories are not necessarily equally correlated with each other. To control for this, we randomly permuted the assignment of categories to components for each ($n = 10$) experiment. We considered several relevant vision architectures, which we all trained with backpropagation: convolutional networks (ConvNets, LeCun et al., 1989) (trained on MNIST) and residual networks (ResNets, He et al., 2016) and Vision Transformers (ViTs, Dosovitskiy et al., 2020) (trained on CIFAR-10).

**Spatial distance of components impacts salience in internal representations.** First, we examined how changes in the input structure impact the networks' representational geometry. We hypothesized that the ConvNets' local weight structure should produce a more conjunctive representation for digits that are closer together. To test this, we determined $S(1; 2)$ in an intermediate layer of the network, averaging over different instances of all digits. We found that $S(1; 2)$ was indeed smaller for lower distances (Fig. 5a), indicating a more conjunctive population. This suggests that varying the distance between two digits provides a practical way of manipulating $S(1; 2)$; below we use this insight to test predictions about how conjunctivity influences generalization behavior.

**Deep networks tend to implement conjunction-wise additive computations.** Conjunction-wise additivity imposes a substantial computational restriction: on test set inputs, models can only add up values assigned to each conjunction of components seen during training. We therefore investigated whether the deep networks' computations could be captured in these terms — a nontrivial prediction as these networks can in principle express arbitrary compositional computations. Remarkably, we found that a conjunction-wise additive model was highly predictive of the model responses (see Appendix D.3). This indicates that large-scale neural networks tend to implement conjunction-wise additive computations, at least when trained on conjunction-wise additive tasks. Our theory suggests that they do so because the statistical structure of the dataset makes such a computation natural.

**Deep networks are impacted by a memorization leak on symbolic addition.** Having found that our theory captures the networks' general computational structure, we investigated whether it was also able to explain their specific performance. We first considered symbolic addition, varying the size of the training set and whether generalization required interpolation or also extrapolation. We trained the ConvNets on 20,000 samples and the ResNets and ViTs on 40,000 samples (Appendix E.1.3).

We tested three theoretical predictions made by Proposition 5.1. First, our theory predicts that the model's test set response should be distorted by a proportional factor. To test this, we plotted the average model prediction for each combination of categories against the ground truth. We found

that model predictions were indeed compressed relative to the ground truth, but still exhibited a strong linear relationship (Fig. 5b). This was also the case for ResNets and ViTs, which exhibited a slightly noisier linear relationship (Fig. 11). The memorization leak therefore affects generalization in large-scale networks as well. This is especially significant as our theoretical argument suggests that memorization leaks likely arise for a broad range of compositional tasks.

We then estimated the slope of this linear dependency for each dataset. Our theory predicts that the compositional generalization slope should increase with increasing $S(1; 2)$ (i.e. increasing distance between components, Fig. 5a). On ConvNets, we found that higher distance indeed increases the slope (Fig. 5c). This was not due to an increase in task difficulty, as in-distribution generalization is not systematically affected by distance. On ResNets and ViTs, higher distance also yielded a higher slope on the compositional generalization set (though the effect was much subtler for ViTs). (Notably, these networks did not perfectly predict the training split, which may affect the results.)

Finally, our theory predicts that larger training sets should increase the compositional generalization slope. Our experiments confirmed this prediction (Fig. 5d). Further, whether the training set required interpolation or also extrapolation did not systematically affect the resulting slope, as predicted by our theory (though interestingly, it did affect in-distribution generalization). Notably, there was one exception to the latter finding: for $p = 4$, the extrapolation dataset had a smaller slope than the interpolation data set. Our theory can therefore explain much but not all of the deep network behavior.

**Deep networks are impacted by a shortcut bias on context dependence.** Lastly, we trained the ConvNets on an MNIST version of context dependence using 30,000 training samples and the ResNets and ViTs on a CIFAR-10 version of the task using 40,000 training samples. Again, their behavior was aligned with the kernel theory's predictions, having better-than-chance accuracy on *CD-1* and *CD-2*, but having worse-than-chance accuracy on *CD-3* (Fig. 5e). This confirms that the deep networks also suffered from a context-driven shortcut on *CD-3* (see also Fig. 16).

# 7 DISCUSSION

Humans often generalize to new situations by stitching together concepts and knowledge from prior experience. Despite the broad importance of this ability (both for cognitive science and machine learning), a general theory of when and how neural networks accomplish compositional generalization has remained elusive. Here we have taken a step towards formalizing this relationship by characterizing kernel models with compositionally structured representations, a framework that captures neural network training or finetuning in the kernel regime and more broadly lets us understand the impact of representational geometry on generalization. We found that they implement a specific compositional computation ("conjunction-wise additivity"). We then investigated how dataset statistics and representational geometry impact successful generalization, highlighting two failure modes arising from the inductive bias of gradient descent (memorization leak and shortcut bias). Finally, we validated our theory in deep neural networks trained on natural image data.

Our results show how simple statistical models can implement (or at least approximate) abstract rules like context dependence. However, they also highlight that building in compositional structure is often insufficient for compositional generalization. Indeed, contextual generalization is highly sensitive to the specific representational geometry and training data. This may explain why investigations into compositional generalization and the benefits of disentangled representations have yielded such inconsistent results. Overall, our insights highlight the utility of kernel models in systematically investigating model generalization. We here used these models to characterize deep neural networks, but they could be equally useful for better understanding human compositional generalization.

While our work covers a broad range of different tasks, a number of limitations remain. We demonstrate that our theory captures many qualitative phenomena in deep neural networks, but do not provide any quantitative bounds. Further, though out of scope here, it would be interesting to test this theory in extremely large-scale models, e.g. large language models. Future work should also consider other input/output formats (e.g. sequences with variable length) and non-compositionally structured representations. Most importantly, our theory is limited to a particular learning mechanism (kernel models). Other learning mechanisms could overcome the limitations highlighted in Sections 4 and 5 (we give an initial example in Appendix F). By helping us design new benchmark tasks not solvable by kernel models, our theory provides an important foundation for such advances.

ACKNOWLEDGMENTS

We are grateful to the members of the Center for Theoretical Neuroscience for helpful comments and discussions. We thank Elom Amematsro, Ching Fang, and Matteo Alleman for detailed feedback, and Zeb Kurth-Nelson for comments on the manuscript. The work was supported by NSF 1707398 (Neuronex), Gatsby Charitable Foundation GAT3708, and the NSF AI Institute for Artificial and Natural Intelligence (ARNI).

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

# A MATHEMATICAL ANALYSIS

## A.1 GENERALIZED COMPOSITIONALLY STRUCTURED REPRESENTATIONS

In the main text, we considered "compositionally structured" representations. This is the class of representations $\phi(z)$ whose kernel $K_\phi(z, z') = \phi(z)^T \phi(z')$ only depends on the number of components $z$ and $z'$ have in common. Here we define a slightly more general class of representations whose kernel depends on the *identity* of the components they have in common, i.e.

$$O(z, z') := \{c = 1, \ldots, C | z_c = z'_c\}. \tag{4}$$

Thus, pairs of representations overlapping in the first component may have a different similarity than pairs of representations overlapping in the second component, but any pair of representations overlapping in the first component still need to have the same similarity. This case captures, for example, cases where one feature is more salient than another and therefore influences the representational similarity more strongly, or where conjunctions between certain components are more saliently represented than conjunctions between other components. For example, perhaps our input consists of two objects with different shapes and colors. In that case, we may wish to represent the conjunction between shape and color of the first object more strongly than the conjunction between the shape of the first object and the color of the second.

Note that we consider the constrained definition of compositionally structured representations in the main text purely for didactical purposes. In particular, our observations in Appendices A.3 and A.4 extend to all generalized compositionally structured representations.

## A.2 WHY ARE WE CONSIDERING COMPOSITIONALLY STRUCTURED REPRESENTATIONS?

So far, we have motivated the concept of compositionally structured representations in terms of the fact that disentangled representations as well as many nonlinear transforms of disentangled representations are compositionally structured. Here we discuss a more conceptual motivation for them: compositionally structured representations guarantee that the only basis for generalization is the compositional nature of the data. This is because we ensure that the model has no knowledge about certain components that may be more similar to each other. Otherwise, the component-wise similarity could serve as a basis for generalization. For example, in transitive equivalence (Fig. 2d), if items that are in the same equivalence class are represented as more similar to each other, that would serve as a basis for generalization on the task that is not compositional in nature. For this reason, we are particularly interested in compositionally structured representations, as they ensure that kernel models are only able to generalize compositionally.

Characterizing compositional generalization in non-compositionally structured representations requires us to meaningfully distinguish between compositional and non-compositional aspects of their generalization. We present an example of such an analysis in Appendix A.5, proving that if the non-compositional representational components are random, the models still generalize in a conjunction-wise additive manner in expectation.

## A.3 NONLINEAR TRANSFORMATIONS AND COMPOSITIONAL STRUCTURE

A broad range of transforms $\phi$ induces a kernel $K_\phi(z, z') = \phi(z)^T \phi(z')$ that only depends on the input similarity $K(z, z') = z^T z'$. In particular (as noted in the main text), this condition is satisfied by the hidden layers and neural tangent kernel of randomly initialized neural networks (in the infinite-width limit) (Han et al., 2022):

**Definition A.1.** Given an input $z \in \mathbb{R}^d$, a network depth $L \geq 2$, and a set of widths $H_1, \ldots, H_L \in \mathbb{N}$, we define a neural network by recursively defining the operation $\phi^{(l)}$ of the $l^{\text{th}}$ layer as

$$a^{(0)} := z, \quad a^{(l)} := \phi^{(l)}(a^{(l-1)}) := \sigma\left(\frac{1}{\sqrt{H_l}}\left(W^{(l)} a^{(l-1)} + b^{(l)}\right)\right), \quad W^{(l)} \in \mathbb{R}^{H_l \times H_{l-1}} \tag{5}$$

where $W^{(l)}$ and $b^{(l)}$ are i.i.d. sampled from a random distribution and $\sigma : \mathbb{R} \to \mathbb{R}$ is a nonlinearity. The complete transform is then given by $\phi := \phi^{(L)} \circ \cdots \circ \phi^{(1)}$.

The infinite-width limit is characterized by $H_1, \ldots, H_{L-1} \to \infty$ (the order of these limits does not matter). This setup includes, in particular, the random feature model investigated by Abbe et al. (2023).

Further, the condition is met by most commonly used kernel functions including any radial basis function that depends on the Euclidean $\ell_2$-norm (e.g. the Gaussian kernel). Any nonlinear transform that meets this condition also conserves compositional structure, i.e. if $x$ is compositionally structured then so is $\phi(x)$. This allows us to derive computational restrictions on this broad range of models.

## A.4 PROOF OF THEOREM 4.2

**Theorem 4.2.** *For any kernel model $f$ with a compositionally structured representation, we can find conjunction-wise functions $f_J : \prod_{c \in J} Z_c \to \mathbb{R}$, where $J \subseteq \{1, \ldots, C\}$, such that for any input $x \in \mathbb{R}^d$ representing components $z \in Z$, the model response is given by*

$$f(x) = \sum_{J \in Conj(z|Z^{train})} f_J(z_J), \quad z_J := (z_c)_{c \in J}. \tag{2}$$

*Proof.* Because the kernel $K$ is compositionally structured, its similarity $K(z, z')$ only depends on the overlap $O(z, z') \subseteq \{1, \ldots, C\}$ (see definition in Eq. (4)). We denote the similarity for inputs overlapping in $S \subseteq \{1, \ldots, C\}$ by $\kappa_S$ and define set of training items overlapping with $z \in Z^{test}$ in $S$ as

$$Z^{train}(z, J) := \left\{ z^{tr} \in Z^{train} | \forall_{c \in J} z_c = z_c^{tr} \right\}. \tag{6}$$

The key idea is to decompose $Z^{train}$ into these different overlaps in order to separate the sum into its components. However, by our definition, the datasets $Z^{train}(z, J)$ are not disjoint. Indeed, $S \subseteq S'$ implies $Z^{train}(z, J) \subseteq Z^{train}(z, J')$ and in particular $Z^{train}(z, \emptyset) = Z^{train}$. To adjust for this, we define $\delta_J$ as the similarity added by $\kappa_S$ to the similarity between conjunctions with one component fewer, recursively defining

$$\delta_\emptyset = \kappa_\emptyset, \quad \delta_J = \kappa_J - \sum_{J' \subsetneq J} \delta_{J'}. \tag{7}$$

We then decompose

$$f(z) = \sum_{z^{tr} \in Z^{train}} a_{z^{tr}} K(z, z^{tr}) = \sum_{J \subseteq \{1, \ldots, C\}} \delta_S \sum_{z^{tr} \in Z^{train}(z, J)} a_{z^{tr}}. \tag{8}$$

This equality obtains because for each $z \in Z, z^{tr} \in Z^{train}$,

$$\sum_{J : z^{tr} \in Z^{train}(z, J)} \delta_J = \delta_{O(z, z^{tr})} + \sum_{J' \subsetneq O(z, z^{tr})} \delta_{J'} = \kappa_{O(z, z^{tr})} = K(z, z^{tr}), \tag{9}$$

which is true by definition. We note that for $J \notin \text{Conj}(z|Z^{train})$, $Z^{train}(z, J) = \emptyset$. Defining

$$f_J(z) := \delta_J \sum_{z^{tr} \in Z^{train}(z, J)} a_{z^{tr}}, \tag{10}$$

proves the proposition. $\square$

## A.5 NON-COMPOSITIONALLY STRUCTURED REPRESENTATIONS

So far, we have considered representations that are compositionally structured. In practice, however, representations will almost never be exactly compositionally structured. Even when there is no specific similarity structure within components, there will be random noise in the representations. Here we characterize one such scenario, demonstrating that in expectation, these representations still yield conjunction-wise additive computations.

**Proposition A.2.** *Consider an input with $C$ components. We consider a representation that represents each component and conjunction of components $z_J$, $J \subseteq \{1, \ldots, C\}$ by a random vector $x_J[z_J] \in \mathbb{R}^d$, $x_J \sim \mathcal{N}(0, \sigma_k^2)$, $\sigma_k^2 > 0$, where $k = |J|$. The representation itself is given by the sum of all these vectors: $x = \sum_{J \subseteq \{1, \ldots, C\}} x_J[z_J]$. All $x_J[z_J]$ are sampled independently from each other. This means that the similarity between different components as well as the entanglement of different components varies randomly. As a result, the kernel regression estimator $f(z)$ is not conjunction-wise additive. However, its expectation, $\mathbb{E}[f(z)]$, is conjunction-wise additive, indicating that all deviations from conjunction-wise additivity arise from random noise and cannot be used for systematic generalization.*

*Proof.* We consider the training representation $X = (x[z_J])_{z \in L^{\text{train}}}$, $X \in \mathbb{R}^{N_{\text{train}} \times d}$ and the test representation $\tilde{X} = (x[z_J])_{z \in L^{\text{test}}}$, $\tilde{X} \in \mathbb{R}^{N_{\text{test}} \times d}$. This gives rise to the training kernel $K = XX^T$ and the train-test kernel, $\tilde{K} = \tilde{X}X^T$. The dual coefficients are given by $a = K^{-1}y$. We now consider a test set input $\tilde{x} = \tilde{x}[\tilde{z}]$ representing the underlying components $\tilde{z}$. Denoting the similarity to the training set by $k(\tilde{x}, X) = X\tilde{x} \in \mathbb{R}^{N_{\text{train}}}$, the test set behavior is given by

$$f(\tilde{x}) = a^T k(\tilde{x}, X) = \sum_{J \in \text{Conj}(\tilde{z}|Z^{\text{train}})} a^T X x_J[z_J] + \sum_{J \notin \text{Conj}(\tilde{z}|Z^{\text{train}})} a^T X x_J[z_J] =: f_1(\tilde{x}) + f_2(\tilde{x}),$$
(11)

where we've simply split up the sum into the conjunction-wise additive part and the remainder. Clearly, this remainder, $f_2(\tilde{x}) := \sum_{J \notin \text{Conj}(\tilde{z}|Z^{\text{train}})} a^T X x_J[z_J]$, will generally not be zero. However, we will now prove that $\mathbb{E}[f_2(\tilde{x})] = 0$. As $f_1(\tilde{x})$ is conjunction-wise additive, this will prove the proposition. To do so, we note that we can assume that we have first sampled all Gaussian vectors relevant for the training set (we will denote this set of random variables by $\mathcal{X}^{\text{train}}$) and subsequently sample the set of Gaussian vectors only relevant for the test trial, i.e. $(\tilde{x}_J)_{J \notin \text{Conj}(\tilde{z}|Z^{\text{train}})}$ (we will denote this set of random variables by $\mathcal{X}^{\text{test}}$). Then,

$$\mathbb{E}[f_2(\tilde{x})] = \mathbb{E}_{\mathcal{X}^{\text{train}}}\left[\mathbb{E}_{\mathcal{X}^{\text{test}}}\left[f_2(\tilde{x})|\mathcal{X}^{\text{train}}\right]\right] = \sum_{J \notin \text{Conj}(\tilde{z}|Z^{\text{train}})} \mathbb{E}_{\mathcal{X}^{\text{train}}}\left[a^T X \mathbb{E}_{\mathcal{Z}^{\text{test}}}\left[x_J[\tilde{z}_J]|\mathcal{Z}^{\text{train}}\right]\right]. \quad (12)$$

This is zero, as $\mathbb{E}_{\mathcal{X}^{\text{test}}}\left[x_J[z_J]\right] = \mathbb{E}_{\mathcal{X}^{\text{test}}}\left[x_J[z_J]|\mathcal{X}^{\text{train}}\right] = 0$, which follows from the fact that $x_J[z_J]$ for $J \notin \text{Conj}(\tilde{z}|Z^{\text{train}})$ is sampled independently from $\mathcal{X}^{\text{train}}$. $\square$

To test our theory empirically, we sample a range of Gaussian representations and train them on symbolic addition, transitive equivalence, and context dependence. For symbolic addition and transitive equivalence, we consider representations with expected $S(1; 2) \in [0.1, 1]$ using ten equally spaced values (i.e. $0.1, 0.2, \ldots$). For context dependence, we consider two types of representation: rep. 1, where $\sigma_1 = 1, \sigma_2 = 0.5, \sigma_3 = 0.1$, (i.e. single components are most saliently represented), and rep. 2, where $\sigma_1 = 0.5, \sigma_2 = 1, \sigma_3 = 0.1$ (i.e. conjunctions of two components are most saliently represented). We consider representations with $d = 100$ and sample 100 instances of each representations. We then fit each representation to these three tasks.

We first estimate these models' additivity on symbolic addition and context dependence. We find that while many model instance are highly additive, some can be highly non-additive (Fig. 6a). However, when averaging across the model behavior 100 randomly sampled representations, this average behavior is perfectly described by a conjunction-wise additive function. This empirically confirms our proposition.

Our proposition implies that these non-compositionally structured representations can still only systematically generalize on conjunction-wise additive tasks. To confirm this insight, we plotted the distribution of accuracies on the transitive equivalence problem across all random model instances. We found that while the models consistently learned the training set, they were indeed unable to generalize to the test set (Fig. 6b).

Finally, we investigated whether our analysis in Section 5 predicted the behavior of randomly sampled representations. Importantly, we have no theoretical guarantees for this scenario. Interestingly, however, the model behaviors were still well predicted by our theory. Specifically, we estimated the slopes that best described the compositional generalization behavior on symbolic addition for different training sets (as in Fig. 5). We found that across different model instances, these slopes were highly varied (Fig. 6c). However, the average slope across all random samples was well predicted by our analytical theory.

Similarly, our theoretical insights on context dependence provided meaningful insight into the generalization behavior of these randomly sampled models. Specifically, on CD-3, the model more saliently representing the single components failed to generalize, whereas the model more saliently representing conjunctions of two components generalized above chance. In contrast, on CD-2, both models generalized above chance. This indicates that our insights on how prevalent shortcut biases are for different training datasets extends to randomly sampled representations.

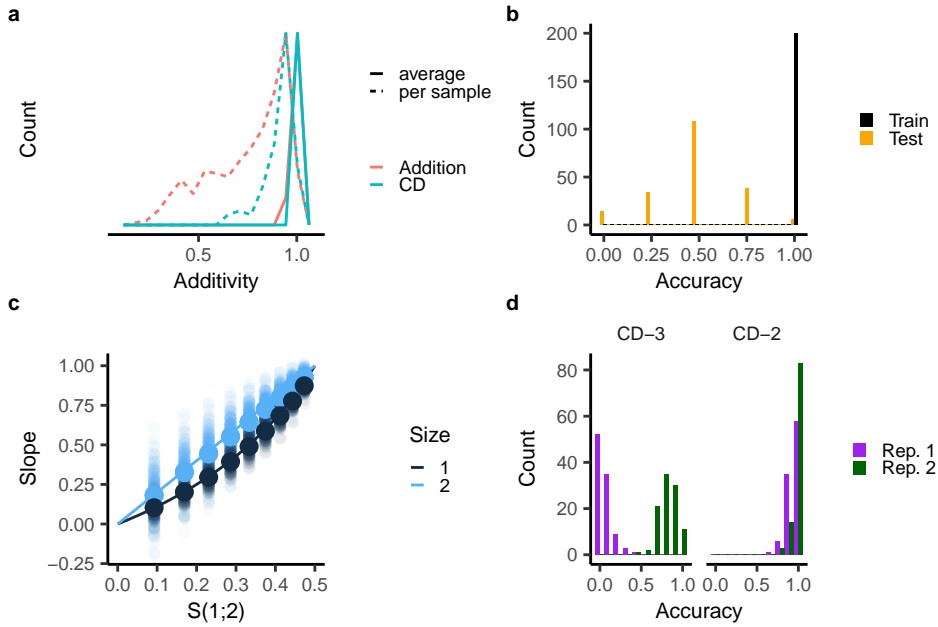

Figure 6: Analysis of randomly sampled representations as considered in Proposition A.2. In all cases we sample 100-dimensional representations. For addition, we consider representations with weights that explore $S(1;2) \in [0.1, 1]$ and for context dependence, we consider representations where $(\sigma_1, \sigma_2, \sigma_3) \in \{(0.5, 1, 0.1), (1, 0.5, 0.1)\}$, where $\sigma_k$ indicates the standard deviation of the Gaussian vector representing conjunctions of $k$ components. **a**, We compare the additivity of individual models to the additivity of the model behavior averaged across 100 randomly sampled representations. While individual models may be severely non-additive, their average behavior is consistently highly additive, empirically confirming our proposition. **b**, As a result, when these models are trained on transitive equivalence, they also systematically exhibit chance performance, as they are still, on average, constrained to be conjunction-wise additive. **c**, We then estimated the slope of the model trained on symbolic addition for each random instance of a representation against corresponding representation's $S(1;2)$. The individual slope estimates are depicted by the small translucent dots, whereas the average across all random instances is depicted by the larger points. Finally, our theoretical predictions in the exactly compositionally structured case are depicted by the lines. The average model behaviors are well described by our theory. **d**, We plot the distribution of generalization accuracies on context dependence across different model seeds, the two considered representations, and for CD-3 and CD-2. On CD-3, the representation with a higher salience for individual components (rep. 1, $(\sigma_1, \sigma_2, \sigma_3) = (0.5, 1, 0.1)$ performs systematically below chance, whereas the representation with a higher salience for conjunctions of two components (rep. 2, $(\sigma_1, \sigma_2, \sigma_3) = (1, 0.5, 0.1)$) generalizes above chance. In contrast, on CD-2, both representation exhibit better-than-chance-accuracy.

## B  REPRESENTATIONAL SALIENCE: A METRIC FOR COMPOSITIONALLY STRUCTURED REPRESENTATIONS

### B.1  DEFINITION

Below we formally define representational salience. To do so, we denote the similarity between two trials $z, z'$ by $\text{Sim}(O(z, z')) := K(z, z')$. Note that this is well-defined for compositionally structured representations, as $K(z, z')$ is identical for all pairs of trials with the same $O(z, z')$. We then define:

**Definition B.1.** For a generalized compositionally structured representation with kernel $K$ and a conjunction $J \subseteq \{1, \ldots, C\}$, we recursively define

$$\overline{S}(\emptyset) := K(\emptyset), \quad \overline{S}(J) := \text{Sim}(J) - \sum_{J' \subsetneq J} \overline{S}(J'), \quad S(J) = \frac{\overline{S}(J)}{\sum_{\emptyset \neq J \subseteq \{1, \ldots, C\}} \overline{S}(J)}. \quad (13)$$

When emphasizing the total number of components, we denote $S(J; C) = S(J)$. Further, for compositionally structured representation, the similarity only depends on the total number of overlaps $|J|$. We therefore write $S(k; C) = S(J)$, where $k := |J|$.

To illustrate how salience would be computed in practice, we consider an example representation $X \in \mathbb{R}^{n \times d}$. This representation gives rise to a kernel $K = XX^T \in \mathbb{R}^{n \times n}$. We can also understand the trial-by-trial similarity in terms of the components it is overlapping in — denoting the set of all subsets of $\{1, \ldots, C\}$ by $\mathcal{J}$, we denote this by $O \in \mathcal{J}^{n \times n}$. In particular, $O_{ii} = \{1, \ldots, C\}$. For example, if data points 1 and 2 overlap in the third component, $O_{12} = \{3\}$. $\text{Sim}(J)$ is then defined as the entries $K_{ij}$ where $O_{ij} = J$. Note that in a compositionally structured representation, whenever $O_{ij} = J$, $K_{ij}$ takes on the same value, but more generally, we can define $\text{Sim}(J)$ by taking the average similarity. We can then define the unnormalized salience $\overline{S}$ by recursively computing it from $\text{Sim}(J)$ using Eq. (13). We then normalize the salience to get our final estimate.

### B.2  WHY IS REPRESENTATIONAL SALIENCE A USEFUL METRIC?

Intuitively, the representational salience captures the unique contribution of a population representing a particular conjunction $J$ (as in our informal definition in the main text). As we noted in the main text, distinguishing these unique contributions from the similarities directly would be more difficult. For example, changes in the similarity between inputs having two components in common could arise from changes in how saliently single components or conjunctions of two components are represented. To distinguish between these cases, we'd have to additionally look at the similarity between inputs having a single component in common. Our definition of representational salience avoids this issue; we therefore believe that this perspective could more broadly be useful for analyzing representations of compositional data.

### B.3  WHY DOES REPRESENTATIONAL SALIENCE LEAVE OUT MAGNITUDE AND BASELINE ACTIVITY?

Notably, many learning models are largely invariant to a constant rescaling of their representation. For limit behavior (e.g. in the limit of infinite training in gradient descent), this scale has no impact. We see this, for instance, in Proposition 5.1, where the scaling turns out to be entirely irrelevant. For regularized models, on the other hand, the scale of the representation is confounded with the strength of the regularization and so we suggest that it is again best seen as separate from the representational geometry.

The baseline activity, on the other hand, may determine how easily an intercept is learned. However, many linear readout models (e.g. support vector machines) do not regularize their intercept, again rendering this parameter entirely irrelevant. Notably, in the case of gradient descent, the magnitude of the baseline activity does influence how easily an intercept is learned. However, the impact of this is often negligible; for example, in Proposition 5.1, we again find that the baseline activity is entirely irrelevant.

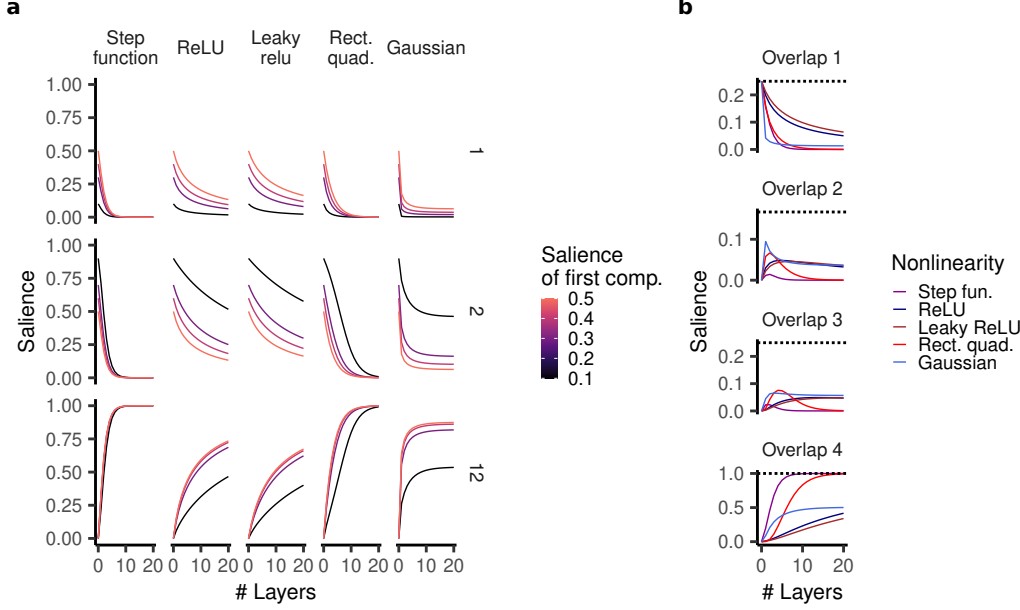

Figure 7: Extended analysis of overlap salience in random neural networks. **a**, Salience of the first component (1), the second component (2), and their conjunction (12), where we vary the two components' saliences in the input. **b**, Salience for inputs with four components.

### B.4 EXTENDED REPRESENTATIONAL ANALYSIS

We computed the saliences by iteratively computing the representational similarities using the kernels derived in prior work (Cho & Saul, 2009; Tsuchida et al., 2018; 2019; Han et al., 2022). In Fig. 7b, we plot the different saliences for an input with four components. Now the salience of overlap 2 and 3 both first increase and then decrease and, just like for inputs with three components (Fig. 3), a Gaussian and rectified quadratic nonlinearity yields a particularly high salience for these intermediate conjunctions. Notably, they appear to be trading off the salience of these conjunctions differently: the rectified quadratic nonlinearity more strongly emphasizes overlaps of three whereas the Gaussian nonlinearity more strongly emphasizes overlaps of two. This highlights that for larger numbers of components, the dependence of generalization behavior on specific architectural choices will likely be even stronger.

Finally, we consider an example of a generalized compositionally structured representation. Here we assume that the different input components have different magnitudes. In particular, we consider a disentangled representation whose first component has a salience between $s \in [0.1, 0.5]$ and whose second component accordingly has a salience $1 - s$ (Fig. 7a). As the random weight neural network becomes deeper, the more salient component (in this case component 2) increasingly dominates the representation. While the salience of the full conjunction still eventually converges to one for most nonlinearities, it takes longer to do so for a less balanced input representation. Further, for a Gaussian nonlinearity, an imbalanced representation actually decreases the limit salience the representation appears to be converging to for the full conjunction. This highlights that for disentangled representations that do not represent all components with equal magnitude, compositional generalization behavior may vary strongly with different neural network architectures.

### B.5 SALIENCE IN VERY DEEP RELU NETWORKS

**Proposition B.2.** *For a random neural network with a (leaky) ReLU nonlinearity, as $L \to \infty$, $S(k; C) \to 0$ for $k < C$ and $S(C; C) \to 1$.*

*Proof.* Note that the proof is a minor extension of Lemma S1.3 in Lippl et al. (2024). We present it here in a self-contained manner. We consider a nonlinearity

$$\sigma(u) := A\min(u,0) + \max(u,0), \quad A \in [0,1]. \tag{14}$$

By prior work (Cho & Saul, 2009; Tsuchida et al., 2018; 2019; Han et al., 2022),

$$\mathbb{E}_w\left[\phi(w^T h^{(l)}(z))\phi(w^T h^{(l)}(z'))\right] = \sigma^2\|h^{(l)}(z)\|_2\|h^{(l)}(z')\|_2 k\left(\hat{h}^{(l)}(z)^T\hat{h}^{(l)}(z')\right), \tag{15}$$

where

$$k(u) = \frac{(1-A)^2}{2\pi}\left(\sqrt{1-u^2} + (\pi - \cos^{-1}(u))u\right) + Au, \tag{16}$$

$\sigma^2$ is the variance of the sampled weights, and $\hat{h} = h/\|\hat{h}\|_2$.

This means that for any two inputs that have a certain similarity $u$, their similarity in the $L$-th layer is given by $k^{(L)}(u)$, where $k^{(L)}$ denotes the $L$-times application of $k$. Let distinct trials in the input have a similarity of $\kappa_d$ and let identical trials have a similarity of $\kappa_i$. Any set of trials with overlapping components will have a similarity $\kappa$, $\kappa_d < \kappa < \kappa_i$. We denote their corresponding similarity in the $L$-th layer by $\kappa_i^{(L)}, \kappa_d^{(L)}, \kappa^{(L)}$. Our goal is now to show that

$$\lim_{L\to\infty} s^{(L)} = 0, \quad s^{(L)} := \frac{\kappa^{(L)} - \kappa_d^{(L)}}{\kappa_i^{(L)} - \kappa_d^{(L)}} = 0. \tag{17}$$

This implies directly that the salience of all partial conjunctions converges to zero, which in turn implies that the salience of the full conjunction converges to one.

(15) implies that $\kappa_i^{(L+1)} = \sigma^2\kappa_i^{(L)}k(1) = \sigma^2\kappa_i^{(L)}\frac{1+A^2}{2}$. Notably, all inputs have the same magnitude and therefore have the same magnitude through all layers; this is given by $\sqrt{\kappa_i^{(L)}}$. We can therefore denote

$$\kappa^{(L+1)} = \sigma^2\kappa_i^{(L)}k\left(\kappa^{(L)}/\kappa_i^{(L)}\right). \tag{18}$$

Thus,

$$s^{(L+1)} = \frac{k(\kappa^{(L)}/\kappa_i^{(L)}) - k(\kappa_d^{(L)}/\kappa_i^{(L)})}{k(1) - k(\kappa_d^{(L)}/\kappa_i^{(L)})} \tag{19}$$

We thus define new normalized variables $\hat{\kappa}^{(L)} := \kappa^{(L)}/\kappa_i^{(L)}, \hat{\kappa}_d^{(L)} := \kappa_d^{(L)}/\kappa_i^{(L)}$, i.e.

$$s^{(L)} = \frac{\hat{\kappa}^{(L)} - \hat{\kappa}_d^{(L)}}{1 - \hat{\kappa}_d^{(L)}}, \tag{20}$$

and therefore

$$\hat{\kappa}^{(L)} = (1 - \hat{\kappa}_d^{(L)})s^{(L)} + \hat{\kappa}_d^{(L)}. \tag{21}$$

Note that $\hat{\kappa}^{(L+1)} = k(\hat{\kappa}^{(L)})/k(1)$ and $\hat{\kappa}_d^{(L+1)} = k(\hat{\kappa}_d^{(L)})/k(1)$. We thus define

$$\tilde{k}(u) := \frac{k(u)}{k(1)} = u + \frac{\rho}{\pi}(\sqrt{1-u^2} - \cos^{-1}(u)u), \quad \rho := \frac{(1-A)^2}{(1+A)^2}. \tag{22}$$

Note that

$$\tilde{k}'(u) = 1 + \frac{\rho}{\pi}\left(-\frac{u}{\sqrt{1-u^2}} - \cos^{-1}(u) + \frac{u}{\sqrt{1-u^2}}\right) = 1 - \frac{\rho}{\pi}\cos^{-1}(u). \tag{23}$$

Note that $k(1) = 1$ and as for all $0 \le u < 1$, $\tilde{k}'(u) < 1$, this is the only fixed point and $\hat{\kappa}^{(L)}, \hat{\kappa}_d^{(L)} \to \infty$. We can therefore define

$$s^{(L+1)} = \frac{\tilde{k}\left((1 - \hat{\kappa}_d^{(L)})s^{(L)} + \hat{\kappa}_d^{(L)}\right) - \hat{\kappa}_d^{(L)}}{1 - \kappa_d^{(L)}}. \tag{24}$$

We now determine the fixed mapping to this mapping assuming that $\hat{\kappa}_d^{(L)}$ is fixed at some value $d$, i.e.:

$$f(s, d) = \frac{\tilde{k}((1 - d)s + d) - \tilde{k}(d)}{1 - \tilde{k}(d)}. \tag{25}$$

$s = 0$ is a fixed point. Further,

$$\frac{\partial f(s, d)}{\partial s} = \frac{(1 - d)\tilde{k}'((1 - d)s + d)}{1 - \tilde{k}(d)} = \frac{(1 - d)(1 - \frac{\rho}{\pi} \cos^{-1}((1 - d)s + d)}{1 - d - \frac{\rho}{\pi}(\sqrt{1 - d^2} - \cos^{-1}(d)d)}. \tag{26}$$

We now prove that this for $0 < s \leq \frac{1}{2}$, this derivative is smaller than 1. Specifically,

$$(1 - d)(1 - \tfrac{\rho}{\pi} \cos^{-1}((1 - d)s + d) =$$
$$1 - d - \tfrac{\rho}{\pi}(\sqrt{1 - d^2} - \cos^{-1}(d)d) + \tfrac{\rho}{\pi}r(s, d), \tag{27}$$

where the residual is given by

$$r(s, d) := (d - 1) \cos^{-1}((1 - d)s + d) + \sqrt{1 - d^2} - d \cos^{-1} d. \tag{28}$$

We now need to prove that $r(s, d) < 0$. Note that $r(s, d)$ is monotonically increasing in $s$ and therefore

$$r(s, d) \leq r(\tfrac{1}{2}, d) = (d - 1) \cos^{-1}(\tfrac{1}{2} + \tfrac{1}{2}d) + \sqrt{1 - d^2} - d \cos^{-1} d < 0, \tag{29}$$

where we infer the latter inequality by visual inspection of the plot of this function. $\qquad\square$

## C  DSPRITES REPRESENTATIONS

To investigate whether our theory can describe compositional generalization in practically used disentangled models, we considered six different model architectures considered in Locatello et al. (2019) and trained on the DSprites dataset, an important benchmark for disentangled representation learning. This paper considers fifty random seeds for each model and further considers six different possible hyperparameters per architectures, resulting in a total of 1,800 models. In our analysis, we only analyze differences between architectures, pooling across all hyperparameter choices.

### C.1  TASK SETUP

DSprites consists of small black-and-white shapes and has five different underlying components: shape (three categories: hearts, ovals, and squares), size, x-position, y-position, and rotation. We consistently consider the largest possible size and hold the rotation fixed at zero. While x- and y-position can each take on 32 possible values, we only consider four different categories for each. Importantly, this is still a highly non-compositionally structured representation: not only do the disentangled representation learning methods often fail to discover the underlying factors of variation; the model also likely represents x- and y-positions that are closer to each other, as more similar, violating our compositionally structured assumption. These representations therefore present a particularly challenging test of our framework.

For symbolic addition and transitive equivalence, we consider x- and y-position as the two components. For each task instance, we randomly determine which position takes on which component's role. In total, we consider 50 randomly sampled task instance for each of the 1,800 models. For context dependence, we subsample two shapes and assume that these shapes provide the context cue. We then consider x- and y-position as feature 1 and feature 2. While the context dependence considered in the main text had six possible feature values for each feature, we now only consider four. As a result, we only consider two different datasets: CD-2, which leaves out all trials where feature 1 indicates category 2 and feature 2 indicates category 1; and CD-1 which only leaves out one conjunction of features. Put differently, on a given trial, the task could for example look as follows: if we see a heart shape, the model should output whether this object is in a certain x-position; if we see a square shape, the model should output whether this object is in a certain y-position. While the model has seen all x- and y-positions individually, it has not seen each combination of x- and y-positions.

Finally, we consider either a direct linear readout from the disentangled representation or an initial transformation by a one-hidden-layer ReLU neural network with random weights.

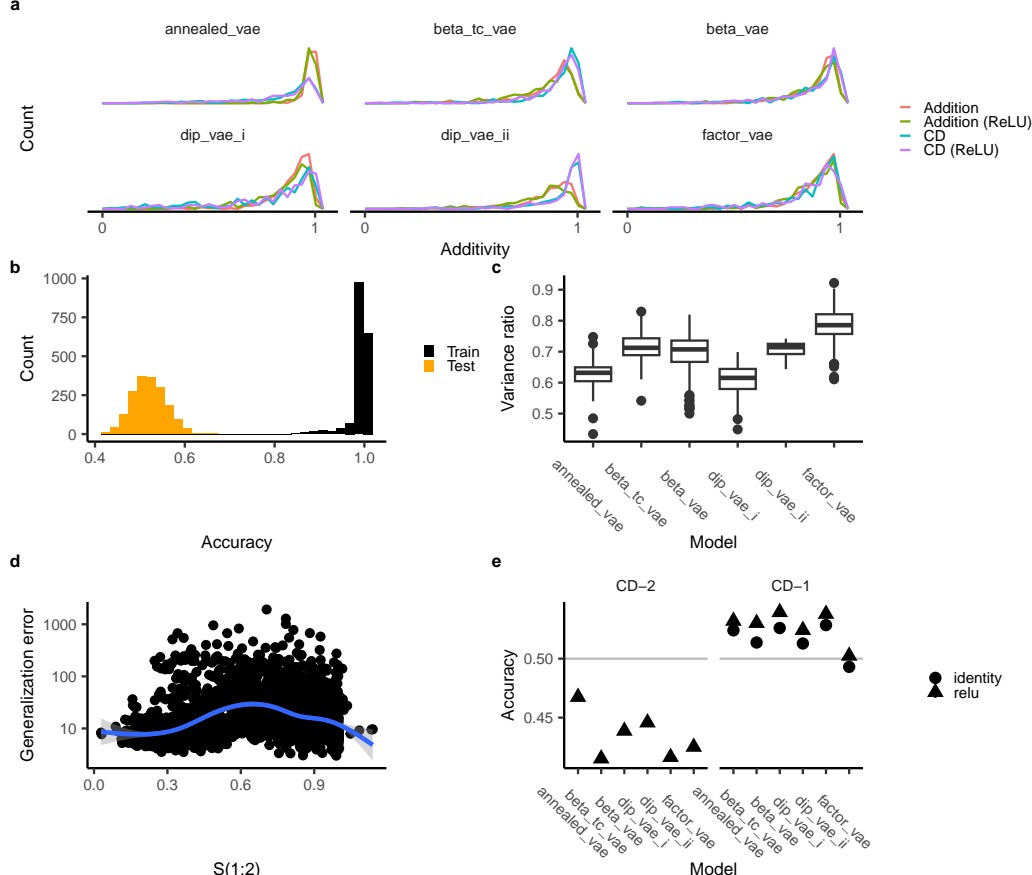

Figure 8: Conjunction-wise additivity in empirical disentangled models. **a**, Additivity across the different model architectures and four different task settings. All of the model behaviors are generally highly additive. **b**, Distribution of accuracies on transitive equivalence when averaged across different component roles. **c**, Variance ratio, capturing the ratio between the deviation between the representational similarity matrix and its compositionally structured approximation and the general variance of the representational similarity matrix. The numbers indicate that all disentangled models discover some compositional structure in their representation, but still deviate substantially from a compositionally structured representation. **d**, Generalization error on symbolic addition plotted against the estimated $S(1;2)$. **e**, Average accuracy on two versions of context dependence with a total of four features: CD-2 which now leaves out an entire orthant, and CD-1, which only leaves out a single data point. Again, models generally perform below chance on CD-2 but above chance on CD-1, though performance is generally really low.

## C.2 THE MODELS ARE WELL DESCRIBED BY A CONJUNCTION-WISE ADDITIVE COMPUTATION

We first investigated whether the model predictions on the test set were well characterized by a conjunction-wise additive computation (see Appendix D.3), averaging this model behavior across the fifty random task instances. We found that they were generally well captured, though they were certainly not perfectly conjunction-wise additive (Fig. 8a). Further, none of these models were able to systematically generalize on transitive equivalence (Fig. 8b). This indicates that conjunction-wise additivity may characterize the generalization class of these highly non-compositionally structured representations as well — at least when averaged across task instances.

## C.3 THE REPRESENTATIONS ARE PARTIALLY COMPOSITIONALLY STRUCTURED

We next investigated whether these representations exhibit compositional structure according to our definition. To determine this, we computed their representational similarity matrix and computed the average similarity for each set of overlaps. This instantiates the compositionally structured representational similarity matrix that most closely described the empirical representational similarity. We then determined the average squared deviation from this matrix, $\sigma_{cs}^2$, comparing it to the overall variance of this matrix, $\sigma_{var}^2$. Overall, the *variance ratio*, $\sigma_{cs}^2/\sigma_{var}^2$, characterizes the degree to which the given representation is compositionally structured. In a fully non-compositionally structured representation, the variance ratio will be roughly one, whereas in a fully compositionally structured representation, it will be zero. For the disentangled representation learning models, we found that this variance ratio took on value between 0.5 and 0.9 (Fig. 8c). Thus, the models' representation partially captured compositional structure, but also contained a lot of non-compositional structure.

## C.4 THE NON-COMPOSITIONAL REPRESENTATIONAL STRUCTURE SUBSTANTIALLY IMPACTS MODEL BEHAVIOR

Finally, we tested whether our representational geometry analysis in Section 5 extended to the DSprites representations. We first estimated their salience $S(1; 2)$ by computing the average representational similarity for each overlap and then computing the salience using Eq. (13). Proposition 5.1 would predict that the generalization error should decrease with increasing $S(1; 2)$. However, we do not observe such a trend (Fig. 8d). This indicates that our representational geometry analysis does not apply to the DSprites representations, due to the way in which they violate the compositional structure assumption.

Finally, we determined generalization accuracy on CD-2 and CD-1. We found that most models performed quite poorly, perhaps owing to the fact that shape is barely represented in these models. Nevertheless, we found that most models performed systematically below chance for CD-2 but systematically above chance for CD-1. Our analysis of context dependence in compositionally structured representations suggests that this may be because these models exploit a context-driven shortcut on CD-2 but not CD-1.

Overall, our analysis therefore paints a nuanced picture of the applicability of our theory to practical disentangled representations. Our findings suggest that these models may still be well approximated by conjunction-wise additive computations and that this generalization class may therefore shed light on fundamental limits to the generalization of linear readout models. At the same time, to understand how these models generalize on conjunction-wise additive tasks, taking into account the specific ways in which they violate the compositional structure assumption is likely important.

# D DETAILED METHODS

## D.1 MODELS

**Kernel model.** We fit the kernel models by hand-specifying the kernel and fitting either a support vector regression or classification using `scikit-learn` (Pedregosa et al., 2011). Note that this is equivalent to using a feature basis with the same resulting similarities. However, hand-specifying these similarities enabled us to easily explore the full range of possible representations.

**Rich and lazy ReLU networks.** All networks were trained with Pytorch and Pytorch Lightning Paszke et al. (2019). We consider ReLU networks with one hidden layer and $H = 1000$ units. We initialize by $\sigma\sqrt{2/H}$, considering $\sigma \in [10^{-6}, 1]$. In particular, when reporting results on rich networks (without further specification), we assume $\sigma = 10^{-6}$. When reporting results on lazy network, we assume $\sigma = 1$.

**Compositional MNIST/CIFAR-10.** We create a compositional version of MNIST and CIFAR-10 by concatenating multiple images along different channels. For example, the input to the MNIST version of the symbolic addition task had two channels, each containing one digit whereas the CIFAR-10 version of the symbolic addition task had six channels, three for each digit. We further varied the distance between the concatenated images either presenting them all on top of each other or presenting them offset by a certain number of pixels. Each image category corresponded to a certain component, where its role as ranodmly sampled for each task instance. The output was still given by a single scalar: the total magnitude for symbolic addition and the two categories for context dependence and transitive equivalence.

**Convolutional neural networks.** We considered networks with four convolutional layers (kernel size is five, two layers have 32 filters, two have 64 filters) and two densely connected layers (with 512 and 1024 units). Each layer is followed by a ReLU nonlinearity, and the convolutional stage is followed by a max pooling operation. All weights are initialized with He initialization (He et al., 2015). We trained these networks with SGD using a learning rate of $10^{-4}$ and momentum of 0.9.

**Residual neural networks.** We trained a residual neural network with eight blocks in total, two with 16, 32, 64, and 128 channels, respectively, using the Adam optimizer with a learning rate of $10^{-3}$ for 100 epochs.

**Vision Transformers.** Finally, we trained a Vision Transformer (ViT) with six attention heads, 256 dimensions for both the attention layer and the MLP, and a depth of four, using Adam with a learning rate of $10^{-4}$ for 200 epochs.

**Data augmentation.** We did not use data augmentation for MNIST. For CIFAR-10, we used a random flip and a random crop.

## D.2 REPRODUCIBILITY

The code required to reproduce all experiments can be found under `https://github.com/sflippl/compositional-generalization`.

## D.3 ADDITIVITY ANALYSIS

To analyze how well a conjunction-wise additive computation can describe network behavior, we considered as the set of possible features a concatenation of one-hot vectors coding for each possible conjunction. We then removed all features that are constant at zero on the training dataset and used linear regression to try and predict network behavior on both training and test set for all remaining features. The resulting $R^2$ defines the "additivity" of the network behavior (i.e. $R^2 = 1$ indicates full conjunction-wise additivity). Furthermore, we can use the inferred values assigned to these different conjunctions to compare kernel models, feature-learning networks, and vision networks. Note that for the vision networks, we first average the model predictions across all different images instantiating a given compositional input.

Notably, our theory predicts that the constrained conjunction-wise additive function can predict behavior on the test set, but not necessarily on the training set. To test this prediction in vision network, we compared the additivity of the test set alone to the additivity of the entire dataset (Fig. 9). First, we found that the predictivity on the test set was indeed generally very high. Second, we found that it was substantially higher than on the training and test set taken together.

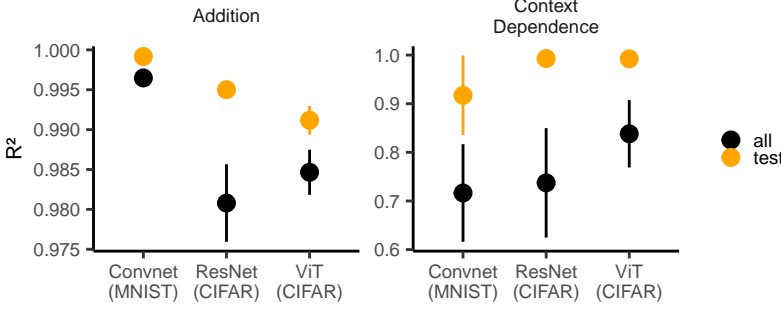

Figure 9: $R^2$ of the conjunction-wise additive predictor on the test set as well as on both the training and test set.

# E COMPOSITIONAL TASK SPACE

## E.1 SYMBOLIC ADDITION

### E.1.1 PROOF OF PROPOSITION 5.1

**Proposition 5.1.** *Consider inputs $z_1, z_2 \in \{[v]\}_{v \in \mathcal{V}}, \mathcal{V} \subset \mathbb{R}$, with associated values $v$. We assume that the training set contains all pairs such that at least one component is $z_c \in \{[w]\}_{w \in \mathcal{W}}, \mathcal{W} \subset \mathcal{V}$ and that the average value in both $\mathcal{V}$ and $\mathcal{W}$ is zero. Then, model behavior on the test set is given by*

$$f([v_1], [v_2]) = m(v_1 + v_2), \quad m := \frac{p \cdot S(1;2)}{1 + (p-2)S(1;2)}, \quad p := |\mathcal{W}| \tag{3}$$

*Proof.* We split up the training data into

$$\mathcal{I}^{(1)} := \{[w] | w \in \mathcal{W}\}^2, \tag{30}$$

and

$$\mathcal{I}^{(2)} := \bigcup_{w \in \mathcal{W}} \mathcal{I}^{(1,w)} \cup \mathcal{I}^{(2,w)}, \tag{31}$$

$$\mathcal{I}^{(1,w)} := \{([w], [v]) | v \in \mathcal{V} \setminus \mathcal{W}\}, \quad \mathcal{I}^{(2,w)} := \{([v], [w]) | v \in \mathcal{V} \setminus \mathcal{W}\}. \tag{32}$$

We denote the dual coefficient associated with each training point $([i], [j])$ by $a_{ij} \in \mathbb{R}$. Note that the problem is symmetric and therefore we know that $a_{ij} = a_{ji}$. We define a few summed coefficients:

$$b_v := \sum_{w \in \mathcal{W}} a_{vw}, \quad \bar{b}_w := \sum_{v \in \mathcal{V} \setminus \mathcal{W}} a_{vw}, \quad c_w := \sum_{w' \in \mathcal{W}} a_{ww'}, \tag{33}$$

$$b := \sum_{v \in \mathcal{V} \setminus \mathcal{W}} b_v = \sum_{w \in \mathcal{W}} b_w, \quad c := \sum_{w \in \mathcal{W}} c_w. \tag{34}$$

Note that the sum over all dual coefficients is given by $2b + c$. Let $p := |\mathcal{W}|$ and $q := |\mathcal{V}| - |\mathcal{W}|$. We denote by $\kappa_c$ the similarity between inputs overlapping in $c$ components. Then, setting $\delta_2 := \kappa_2 - \kappa_0$ and $\delta_1 := \kappa_1 - \kappa_0$, the set of dual equations is given by

$$([w], [w']) \in \mathcal{I}^{(1)} : \kappa_0(2b + c) + \delta_1(\bar{b}_w + \bar{b}_{w'} + c_w + c_{w'}) + (\delta_2 - 2\delta_1)a_{ww'} = w + w', \tag{35}$$

$$([w], [v]) \in \mathcal{I}^{(1,w)} : \kappa_0(2b + c) + \delta_1(b_v + \bar{b}_w + c_w) + (\delta_2 - 2\delta_1)a_{wv} = w + v. \tag{36}$$

(Note that the equation corresponding to $([v], [w])$ is equivalent due to the problem's symmetry.)

The prediction is given by

$$f([v_1], [v_2]) = \kappa_0(2b + c) + \delta_1(b_{v_1} + b_{v_2}). \tag{37}$$

We now sum (36) over $w$ (setting $\overline{w} := \sum_{w \in \mathcal{W}} w$):

$$\begin{aligned}
\overline{w} + pv &= p\kappa_0(2b + c) + p\delta_1 b_v + \delta_1(b + c) + (\delta_2 - 2\delta_1)b_v \\
&= ((p-2)\delta_1 + \delta_2)b_v + (2p\kappa_0 + \delta_1)b + (p\kappa_0 + \delta_1)c
\end{aligned} \tag{38}$$

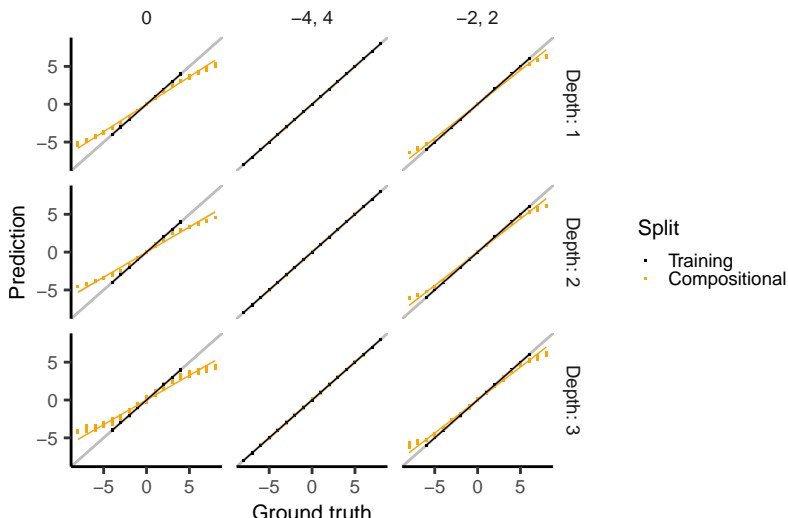

Figure 10: Model prediction plotted against ground truth for different training sets and depths.

Thus,

$$\delta_1 b_v = \frac{\delta_1 p}{(p-2)\delta_1 + \delta_2} v + \delta_1(\overline{w} - (2p\kappa_0 + \delta_1)b - (p\kappa_0 + \delta_1)c). \tag{39}$$

Thus, setting

$$m := \frac{\delta_1 p}{(p-2)\delta_1 + \delta_2}, \quad d := 2\delta_1(\overline{w} - (2p\kappa_0 + \delta_1)b - (p\kappa_0 + \delta_1)c) + \kappa_0(2b + c), \tag{40}$$

we can write

$$f([v_1], [v_2]) = m(v_1 + v_2) + d. \tag{41}$$

To simplify $d$, we sum (35) over all $w, w'$:

$$2p\overline{w} = p^2\kappa_0(2b + c) + 2p\delta_1(b + c) + (\delta_2 - 2\delta_1)c$$
$$= 2p(p\kappa_0 + \delta_1)b + (p^2\kappa_0 + 2(p-1)\delta_1 + \delta_2)c. \tag{42}$$

We further sum (38) over all $v$, setting $\overline{v} := \sum_{v \in \mathcal{V} \setminus \mathcal{W}} v$:

$$((p + q - 2)\delta_1 + \delta_2 + 2pq\kappa_0)b + q(p\kappa_0 + \delta_1)c = q\overline{w} + p\overline{v}. \tag{43}$$

We can now compute $b$ and $c$ from this system of equations and plug it into $d$.

Finally, $\overline{w} = \overline{v} = 0$ immediately implies that $b = c = 0$ and therefore $d = 0$. $\qquad\square$

### E.1.2 BEHAVIOR OF DEEP ReLU NETWORKS

Next, we examined the behavior of deep ReLU networks with varying depth (Fig. 10). On the extrapolation task ($\mathcal{W} = \{0\}$), these networks also had compressed model predictions on the compositional split that were approximately linearly distorted (though for deeper networks, this relationship seemed to have a slight S-shape). Intriguingly, these networks perfectly generalized on the interpolation task ($\mathcal{W} = \{-4, 4\}$). For $\mathcal{W} = \{-2, 2\}$, they perfectly generalized on the interpolation portion, but not the extrapolation portion.

### E.1.3 BEHAVIOR OF VISION MODELS

We trained the Convnets on MNIST using seven different training sets $\mathcal{W}$: $\{[0]\}$, $\{[-4], [4]\}$, $\{[-2], [2]\}$, $\{[-4], [0], [4]\}$, $\{[-1], [0], [1]\}$, $\{[-4], [-3], [3], [4]\}$, $\{[-2], [-1], [1], [2]\}$. This allowed us to vary both the size of the training set and whether it involved only interpolation or also extrapolation. We trained these networks for 100 epochs on 20,000 samples. We trained the ResNets and ViTs on CIFAR-10 using the first three training sets listed above. We trained the ResNets for 100

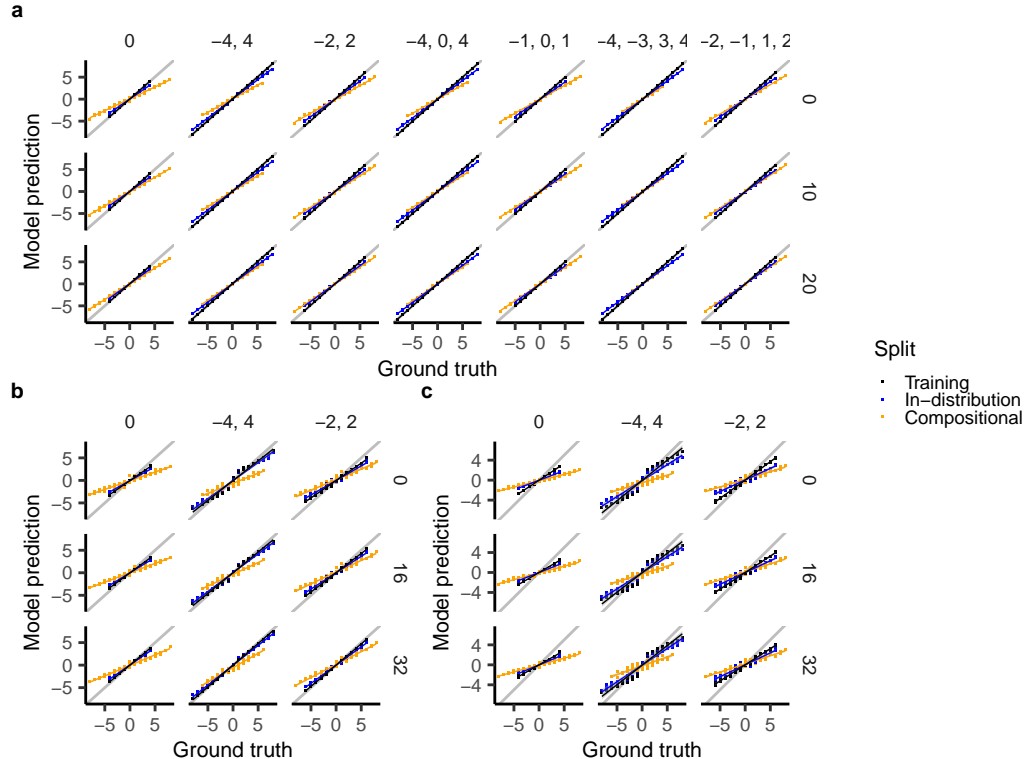

Figure 11: Average prediction for each combination of components plotted against the ground truth for **a**, ConvNets trained on MNIST, **b**, ResNets trained on CIFAR-10, and **c**, ViTs trained on CIFAR-10.

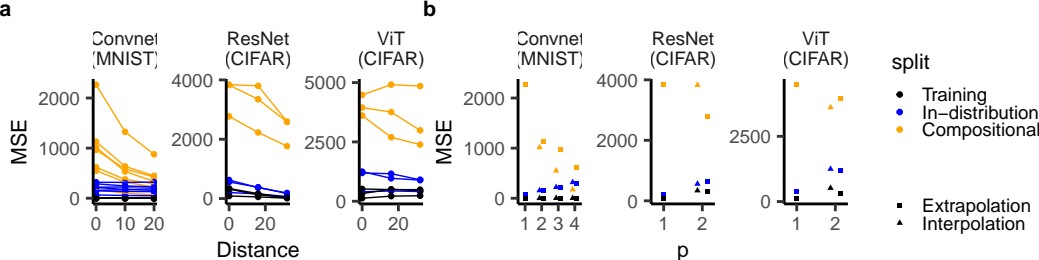

Figure 12: **a**, Mean squared error as a function of distance. **b**, Mean squared error as a function of training set size $p$ (different dots corresponding to interpolation versus extrapolation.

epochs and the ViTs for 200 epochs, training all networks on 40,000 samples. Our findings on these experiments are summarized in Section 6. Further, Fig. 11 depicts, for each training set, the average prediction for each combination of components. We can see that the ConvNet predictions are highly linearly correlated with the ground truth. The ResNet and ViT predictions are also linearly correlated, but exhibit a slightly noisier relationship.

Additionally, we also plot the mean squared error of the networks on the different training sets (Fig. 12). This provides a more conventional (but harder to interpret) measure of performance compared to the slope depicted in Fig. 5. We again see that the mean squared error on the compositional generalization set is decreasing with increasing distance (Fig. 12a). Further, it is decreasing with increasing training set size (Fig. 12), though we note that the differences in the scales of the generalization set for different sets $\mathcal{W}$ renders it harder to directly to compare these values. In contrast, the slopes described in the main text are more easily comparable.

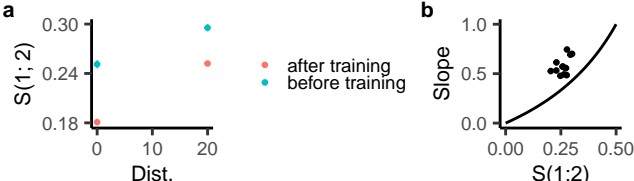

Figure 13: **a**, $S(1; 2)$ in the convolutional neural network's neural tangent kernel, before and after being trained on symbolic addition with $\mathcal{W} = \{0\}$. **b**, The estimated slope for each model instance plotted against $S(1; 2)$. While increased $S(1; 2)$ generally yields a larger slope (as predicted by our theory), the relationship between $S(1; 2)$ and the slope does not follow the exact quantitative relationship predicted by our theory.

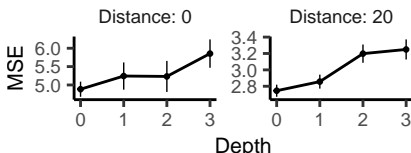

Figure 14: Mean squared error on the compositional generalization set for different depths of the fully connected network at the end of the convolutional neural network architecture.

Overall, our analysis indicates that our theory provides several valuable qualitative insights into the behavior of deep neural network models. Importantly, however, we are not able to provide quantitative bounds at the moment. To demonstrate this, we computed the salience $S(1; 2)$ of the neural tangent kernel of the convolutional neural networks before and after being trained on the MNIST version of symbolic addition with $\mathcal{W} = \{0\}$ (Fig. 13a). Importantly, $S(1; 2)$ changed substantially during training, indicating that these networks are not trained in the kernel regime. As such, our theory does not apply exactly. To see whether it provided appropriate quantitative bounds, we then estimated the slope of each model instance on the compositional generalization dataset and plotted it against $S(1; 2)$ of the neural tangent kernel at the beginning of training. We found that higher $S(1; 2)$ generally resulted in a larger slope, as predicted by our theory. However, the relationship between $S(1; 2)$ and slope did not adhere exactly to the quantitative predictions made by our theory. This indicates that while our theory can shed light on certain qualitative behaviors, it still leaves certain generalization behaviors in deep neural networks unclear.

Finally, our analysis in Section 5.1 suggests that deeper models should yield more conjunctive representations, and therefore exhibit worse generalization on symbolic addition. To test this prediction, we varied the number of fully connected layers in the convolutional neural network and trained these networks on the symbolic addition task with the training set $\mathcal{W} = \{0\}$. We found that deeper networks indeed generalized worse (Fig. 14).

### E.2 CONTEXT DEPENDENCE

#### E.2.1 GENERAL TASK DEFINITION

We consider inputs with three components, $(z_{co}, z_{f1}, z_{f2})$. We assume that $z_{co} \in C_1 \cup C_2$, where $C_1$ is the set of possible contexts under which $z_{f1}$ is relevant and $C_2$ is the set of possible contexts under which $z_{f2}$ is relevant. We further assume that there are decision functions $d_1(z_{f1}), d_2(z_{f2}) \in \mathbb{R}$. (For example, in the example in the main text, these function map three features to the first category (i.e. $y = -1$) and three features to the second category (i.e. $y = 1$).) The target is then given by

$$y(z_{co}, z_{f1}, z_{f2}) = \begin{cases} d_1(z_{f1}) & \text{if } z_{co} \in C_1, \\ d_2(z_{f2}) & \text{if } z_{co} \in C_2. \end{cases} \tag{44}$$

Note that in the main text, we consider $C_1 = \{1\}$, $C_2 = \{2\}$, and six possible values for $z_{f1}, z_{f2}$, where the decision function maps three onto 1 and three onto $-1$.

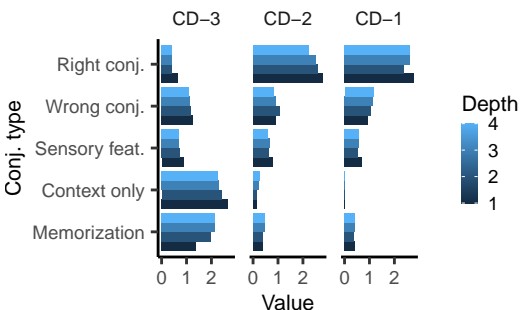

Figure 15: Inferred values for different conjunctive groups on context dependence.

### E.2.2 NOVEL STIMULUS COMPOSITIONS ARE CONJUNCTION-WISE ADDITIVE

If the test set consists in novel combinations of stimuli, this is a conjunction-wise additive computation. Namely, suppose that for all test inputs $(z_{co}, z_{f1}, z_{f2})$, the two features have never been observed in conjunction, but both $(z_{co}, z_{f1})$ and $(z_{co}, z_{f2})$ have been. (This includes the case considered in the main text.) In this case, we can define functions $f_{12}$ and $f_{13}$ to implement the appropriate mapping:

$$f_{12}(z_{co}, z_{f1}) := \begin{cases} d_1(z_{f1}) & \text{if } z_{co} \in C_1, \\ 0 & \text{if } z_{co} \in C_2, \end{cases} \quad f_{13}(z_{co}, z_{f2}) := \begin{cases} 0 & \text{if } z_{co} \in C_1, \\ d_2(z_{f2}) & \text{if } z_{co} \in C_2, \end{cases} \quad (45)$$

$$f(z_{co}, z_{f1}, z_{f2}) = f_{12}(z_{co}, z_{f1}) + f_{13}(z_{co}, z_{f2}). \quad (46)$$

### E.2.3 NOVEL RULE COMPOSITIONS ARE NOT CONJUNCTION-WISE ADDITIVE

We could also imagine an alternative generalization rule in a task where there are multiple components indicating the same context: $C_1 = \{co_1, co_2\}$ and $C_2 = \{co_3, co_4\}$. We then leave out certain features with certain contexts. For example, suppose we had never seen two values for $z_{f1}$ and $z_{f2}$ in conjunction with $z_{co} \in \{co_2, co_4\}$. In principle, if the model understood that $z_{co} = co_1, co_2$ (and $z_{co} = co3, co4$ resp.) signify the same context (i.e. learned to abstract the context from the context cue), it could generalize successfully as it had observed these features in conjunction with $z_{co} = co_1, co_3$. However, the conjunction-wise additive mapping depends on having observed each context in conjunction with each feature and this task is therefore non-additive.

### E.2.4 COEFFICIENT GROUPS

In Fig. 4d, we grouped the inferred coefficients into categories. We here explain these categories:

- **Right conj.**: This is the correct conjunction the model should use to solve the task, i.e. between $z_{co} = co_1$ and $z_{f1}$ and between $z_{co} = co_2$ and $z_{f2}$.

- **Wrong conj.**: This is the incorrect conjunction between context and feature, i.e. between $z_{co} = co_1$ and $z_{f2}$ and between $z_{co} = co_2$ and $z_{f1}$.

- **Sensory feat.**: This is any conjunction involving sensory features, i.e. $z_{f1}, z_{f2}, (z_{f1}, z_{f2})$.

- **Context only**: This is the component $z_{co}$ by itself.

- **Memorization**: This is the full conjunction of all three components $(z_{co}, z_{f1}, z_{f2})$.

We then compute the average absolute magnitude within each of these groups in order to determine their overall relevance to model behavior.

### E.2.5 FEATURE-LEARNING RELU NETWORKS

We find that feature-learning ReLU networks generalize consistently on *CD-1* and CD-2 but not *CD-3*. They are also perfectly conjunction-wise additive and fail due to a context shortcut (Fig. 15).

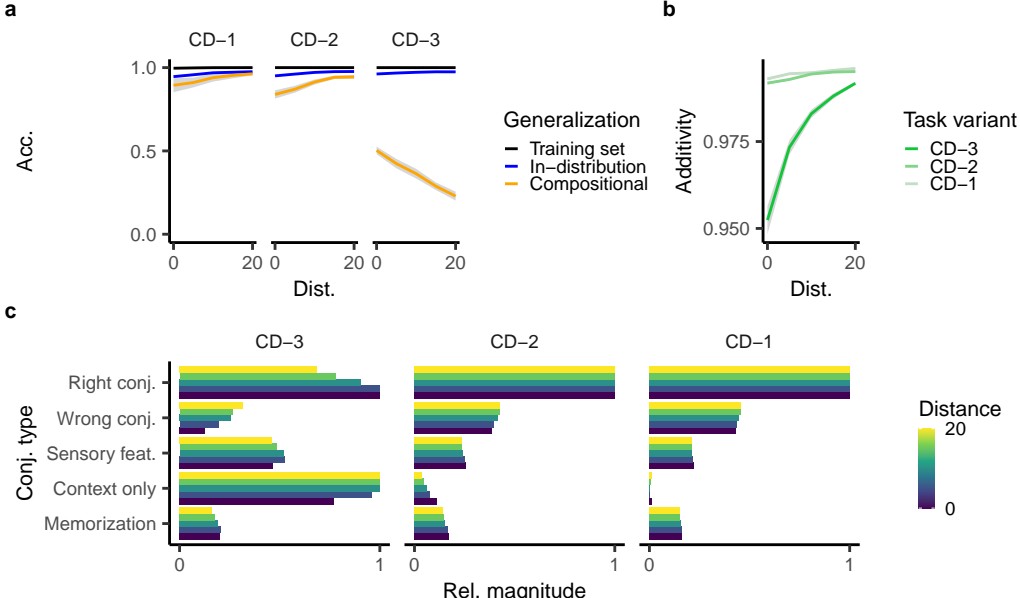

Figure 16: Performance of convolutional neural networks trained on an MNIST version of context dependence. For different distances and training sets, we plot **a**, the accuracy on different splits, **b**, the additivity of the networks, and **c**, the magnitude of the different inferred coefficients.

### E.2.6 VISION NETWORKS

We additionally analyzed ConvNets for different distances between the digits. We found that convolutional neural networks trained on MNIST successfully generalized on *CD-1* and *CD-2*, but not CD-3 (Fig. 16). For smaller distances between digits, the models tended to generalize worse on *CD-1* and *CD-2* and gradually reverted to chance accuracy (i.e. 0.5) for *CD-3*. Further, the networks were generally highly additive ($R^2 > 0.95$), but became worse for lower distance (Fig. 16b). Finally, across all distances, they had a high magnitude associated with the context cue, though this magnitude decreased for small distances — consistent with the accuracy of the network increasing from below chance to chance level (Fig. 16c).

### E.3 INVARIANCE AND PARTIAL EXPOSURE

We consider invariance and partial exposure as simple case studies for the memorization leak and shortcut bias. In both cases, the input consists of two components and the mapping only depends on the first (Fig. 17a). In the invariance case, we don't see the second component vary at all, in the partial exposure case, we see one instance of the second component. Note that the partial exposure task has previously been studied in the context of network generalization (Dasgupta et al., 2022; Chan et al., 2022).

To understand the impact of different representational saliences, we consider the generalization margin $m := y\hat{y}$ on the test set, where $y$ is the ground-truth label and $\hat{y}$ is the model's estimate. Because we consider support vector machines, the margin on the training set is one; a smaller margin on the test set indicates worse performance. We determined a mathematical formula for the margins as a function of $S(1;2)$. Below we first describe its implications and then how we derived this.

**Invariance suffers from a memorization leak.** On invariance, we find that the model's test margin is expressed as $m = \frac{S(1;2)}{1-S(1;2)}$. This means that for a fully compositional representation ($S(1;2) = 0.5$), its training and test margins are both one. However, as $S(1;2)$ decreases, the model increasingly memorizes the training set, resulting in a decreased margin (Fig. 17b).

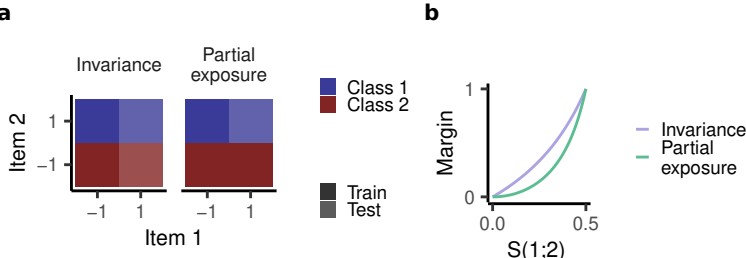

Figure 17: **a**, Schema for invariance and partial exposure. **b**, Generalization margin for the two tasks.

**Shortcut distortion.** On the partial exposure task, if the model used item 1 to solve the task, it would get two out of three training examples correct and could memorize the last data point. This is a statistical shortcut and we find that norm minimization (just as for context dependence) ends up partially relying on this strategy as this decreases the $\ell_2$-norm of the readout weights. As a result, the test margin for the partial exposure task decreases even more strongly as a function of $S(1;2)$: $m = \frac{2S(1;2)^2}{1-2S(1;2)^2}$ (where we assume that the similarity between identical trials is one and the similarity between distinct trials is zero) (Fig. 17b).

**Derivations.** We analytically compute the kernel models' test set prediction on the invariance task. The training set is given by $\{(-1,-1),(-1,1)\}$ and its kernel is therefore

$$K = \begin{pmatrix} \kappa_2 & \kappa_1 \\ \kappa_1 & \kappa_2 \end{pmatrix}, \tag{47}$$

where $\kappa_2$ is the similarity between identical trials and $\kappa_1$ is the similarity between overlapping trials. Hence, the dual coefficients are given by

$$a = K^{-1}\begin{pmatrix} 1 \\ -1 \end{pmatrix} = \frac{1}{\kappa_2^2 - \kappa_1^2}\begin{pmatrix} \kappa_2 & -\kappa_1 \\ -\kappa_1 & \kappa_2 \end{pmatrix}\begin{pmatrix} 1 \\ -1 \end{pmatrix} = \frac{1}{\kappa_2^2 - \kappa_1^2}\begin{pmatrix} \kappa_2 + \kappa_1 \\ -(\kappa_2 + \kappa_1) \end{pmatrix}. \tag{48}$$

The test set is given by $\{(1,-1),(1,1)\}$ and its kernel with respect to the training set is therefore

$$\tilde{K} = \begin{pmatrix} \kappa_1 & \kappa_0 \\ \kappa_0 & \kappa_1 \end{pmatrix}, \tag{49}$$

where $\kappa_0$ is the similarity between distinct trials. Hence the test set predictions are given by

$$\hat{y} = \tilde{K}a = \frac{1}{\kappa_2^2 - \kappa_1^2}\begin{pmatrix} (\kappa_2 + \kappa_1)(\kappa_1 - \kappa_0) \\ -(\kappa_2 + \kappa_1)(\kappa_1 - \kappa_0). \end{pmatrix} \tag{50}$$

As the ground truth labels are $y = \{1, -1\}$, the margin $m = y\hat{y}$ is identical for both test set points:

$$m = \frac{(\kappa_2 + \kappa_1)(\kappa_1 - \kappa_0)}{\kappa_2^2 - \kappa_1^2} = \frac{\kappa_1 - \kappa_0}{\kappa_2 - \kappa_1} = \frac{(\kappa_2 - \kappa_0)S(1;2)}{(\kappa_2 - \kappa_0) - (\kappa_1 - \kappa_0)} = \frac{S(1;2)}{1 - S(1;2)}. \tag{51}$$

For partial exposure, the training set is given by $\{(-1,-1),(-1,1),(1,-1)\}$ and its kernel is therefore

$$K = \begin{pmatrix} \kappa_2 & \kappa_1 & \kappa_1 \\ \kappa_1 & \kappa_2 & \kappa_0 \\ \kappa_1 & \kappa_0 & \kappa_2 \end{pmatrix}. \tag{52}$$

The test set is given by $\{(1,1)\}$ and the test set kernel is therefore

$$\tilde{K} = \begin{pmatrix} \kappa_0 & \kappa_1 & \kappa_1 \end{pmatrix} \tag{53}$$

The margin is therefore given by

$$m = y\hat{y} = -\hat{y} = -\tilde{K}K^{-1}\begin{pmatrix} 1 \\ -1 \\ 1 \end{pmatrix}. \tag{54}$$

We solve this equation for the special case where $\kappa_0 = 0$ and $\kappa_1 = 1$ using Mathematica and find that

$$m = \frac{2S(1;2)^2}{1 - 2S(1;2)^2}. \tag{55}$$

Figure 18: Feature learning enables generalization on transitive equivalence. **a**, Learning curves in the lazy and rich regime. **b**, The weights of the network in the subspace corresponding to one underlying *XOR*-task. **c**, Weights for the same unit are plotted against each other and colored by whether they correspond to equivalent items (purple) or non-equivalent items (green). **d**, Accuracy of a ConvNet trained on an MNIST version of transitive equivalence.

### E.4 OTHER MATHEMATICAL OPERATIONS

We could consider mathematical operations other than addition as well, considering unobserved assigned values $v_1(z_1)$ and $v_2(z_2)$ together with some composition function $C(v_1(z_1), v_2(z_2))$. This task will only be additive if the composition function is additive (e.g. if it is subtraction). If it is, e.g. multiplication, division, or exponentiation, the task will be non-additive.

### E.5 LOGICAL OPERATIONS

In this task, inputs with two components are presented. Each component $z_c$ has an unobserved truth value $T(z_c)$ associated with it and the target is some logical operation over these two truth values, for example *AND*: $T(z_1) \land T(z_2)$. After inferring the truth value of each component, the model could generalize towards novel item combinations. As long as the logical operation is additive (e.g. *AND*, *OR*, *NEITHER*, ...), this is an additive task. If the logical operation is non-additive (e.g. *XOR*), this would be a non-additive task. Indeed, the *XOR* case is structurally equivalent to the transitive equivalence task.

### E.6 TRANSITIVE ORDERING

Transitive ordering is a popular task in cognitive science (often called transitive inference, McGonigle & Chalmers (1977)). Here the subject is presented with two items $z_1$, $z_2$ drawn from an unobserved hierarchy $>$. It should then categorize whether $z_1 > z_2$ or $z_2 > z_1$. Crucially, this task can be solved by assigning a rank $r(z_c)$ to each item and computing the response as $f(z) = r(z_1) - r(z_2)$ Lippl et al. (2024). It is therefore additive, in contrast to transitive equivalence. This is also the case if we assume that there are multiple such hierarchies (e.g. $a_1 > \ldots, a_5$ and $b_1 > \ldots, b_5$). In this case, the model would generalize to comparisons between these different hierarchies as well.

## F  A FEATURE-LEARNING MECHANISM FOR NON-ADDITIVE COMPOSITIONAL GENERALIZATION

We proved above that compositionally structured kernel models do not generalize on non-additive tasks, including transitive equivalence. To see if feature learning can overcome this limitation, we trained ReLU networks through backpropagation on transitive equivalence, using a disentangled and uncorrelated input. By varying the initial weight magnitude, we either trained these networks in the kernel/lazy regime or the feature-learning/rich regime. Notably, when trained on symbolic addition and context dependence, the rich networks were well-described by a conjunction-wise additive model (Figs. 10 and 15). On transitive equivalence, however, while the kernel-regime models failed to generalize (as predicted by our theory), the rich networks generalized correctly (Fig. 18a).

To explain why this is the case, we leveraged the insight that rich neural networks are biased to learning weights with a low overall $\ell_2$-norm (Lyu & Li (2020); Chizat & Bach (2020); cf. Vardi & Shamir (2021)). In particular, a one-hidden-layer ReLU network tends to learn a sparse set of features Savarese et al. (2019); Chizat & Bach (2020). Transitive equivalence consists of multiple overlapping equality relations (e.g. $A = B \neq D = E$). Notably, ReLU networks such an *XOR*-type

problem by specializing one unit to each conjunction (Brutzkus & Globerson (2019); Saxe et al. (2022); Xu et al. (2023); Fig. 18b). Further, their sparse inductive bias incentivizes ReLU networks to use identical units for overlapping conjunctions (e.g. $(A, B), (A, C)$, and $(B, C)$). This causes the unit to generalize to unseen item combinations (e.g. $(A, C)$), enabling the network to generalize. Importantly, our theoretical argument is corroborated by empirical simulations: each network unit has identical weights for equivalent items (Fig. 18c).

Thus, rich networks' capacity for abstraction gives rise to an additional compositional motif, allowing them to generalize on transitive equivalence. In particular, our findings highlight that transitive equivalence and transitive ordering are solved by fundamentally different network motifs.

To see whether large-scale neural networks can also benefit from this feature-learning mechanism, we trained ConvNets on an MNIST version of transitive equivalence. The networks were trained for 150 epochs on 20,000 samples. We found that if the digits were presented with a distance of zero, the network did not generalize compositionally at all. However, with increasing distance, the network started to improve its compositional generalization (Fig. 18d), demonstrating that a convolutional network can benefit from this rich compositional motif.

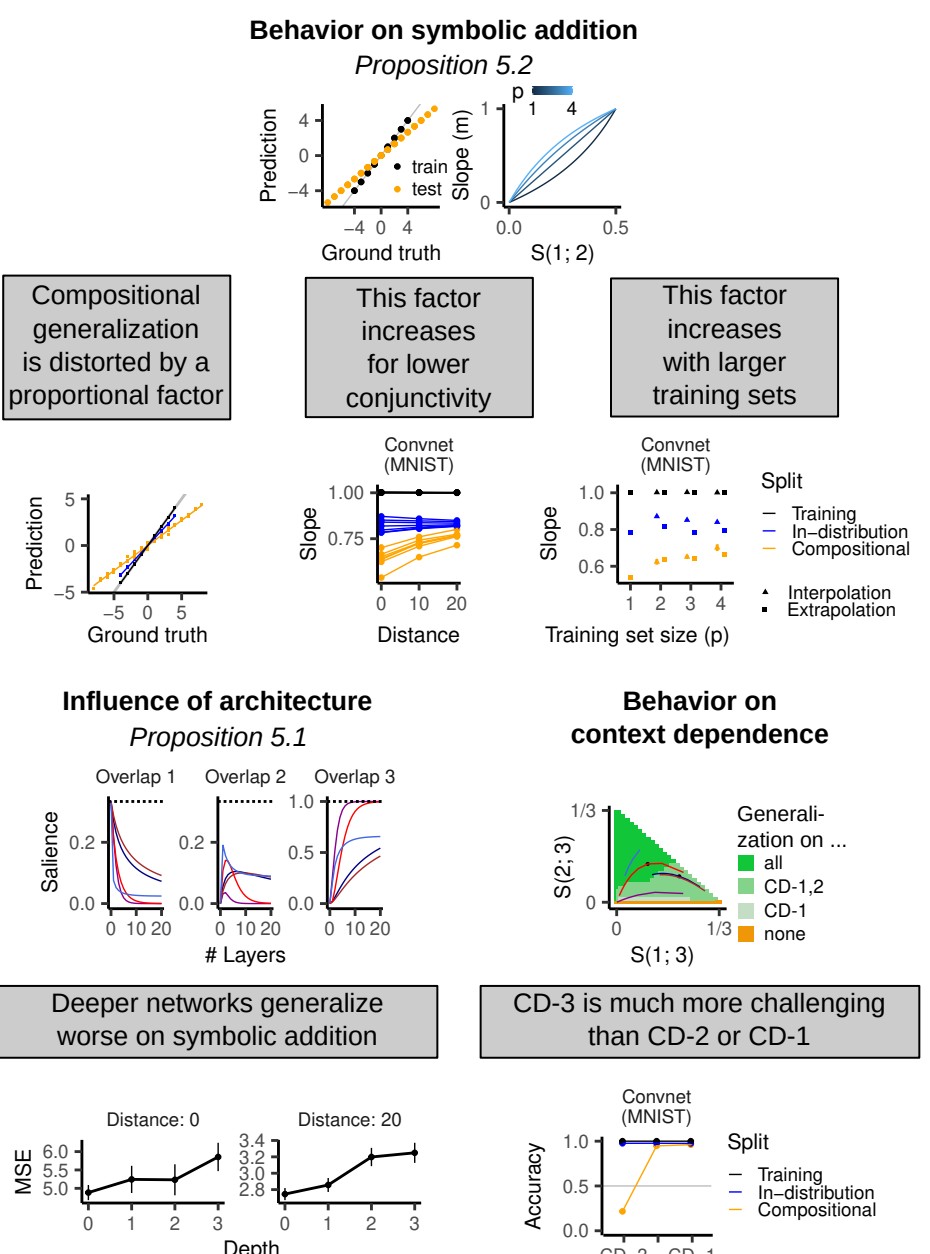

Figure 19: This figure composes the relevant theoretical insights and empirical experiments to illustrate how they speak to each other.

