# OpenReview forum: "When does compositional structure yield compositional generalization? A kernel theory."
_ICLR.cc/2025/Conference — ICLR 2025 Poster_

### Official Review · Reviewer_CS3X · 2024-10-21

**Soundness:** 4
**Presentation:** 3
**Contribution:** 3
**Rating:** 10
**Confidence:** 4

**Summary:**

This paper considers under what circumstances a kernel model learning on top of a disentangled representation can generalize compositionally. The main finding is that these models can generalize compositionally if the task is what the authors call conjunction-wise additive with respect to the disentangled features. Although the theory is developed using kernel models, most of the results generalize more or less to deep neural networks, which in some training regimes are related to kernel models. Overall I really liked this paper. The results are interesting and novel, the paper is for the most part clear and well-written, and the results help push the field forward in terms of designing models which are more likely to generalize compositionally.

**Strengths:**

1. The reason for looking at kernel models is very clearly articulated and motivated.
2. The theory is supplemented with good empirical experiments that show it applies in deep neural networks, even though it was developed with kernel models (with a nice method for deep learn-afying the main tasks used in the kernel models sections).
3. Contributions and advantages over prior work are very clear (e.g., no assumptions about modularity or linear/additive combinations in the downstream model).
4. The practical consequences of the main finding, laid out in Section 4.3, are clearly articulated with several simple examples.
5. The work helps guide the design of more difficult compositional generalization benchmarks that cannot be solved by kernel models (or neural networks trained in the kernel regime).
6. Figures are clear and aesthetic, both when illustrating the framework/tasks and when presenting results.
    1. One exception I recommend fixing is Figure 3d, where the borders of the bar are too thick and obscure the colors of the bars.

**Weaknesses:**

1. The paper discusses compositional generalization with respect to compositional representations, but equates compositional representations with disentangled representations. I think it’s fair to say that other compositional structures are possible in additional to disentanglement (e.g., Tensor product representations, block-wise disentanglement, etc.), and disentanglement is likely just one particular special case of compositional representations. Ideally I would want to see the whole paper rewritten in terms of disentanglement instead of compositionality (disentangled representations, generalization from disentangled representations). I understand that the authors might not want to go this far, so in this case I would make it very clear in the introduction that you are only considering disentangled representations, and that these are a subset of what might be considered compositional representations more generally.
2. At times it is unclear whether this work is about compositional tasks (tasks where the optimal solution would necessitate some compositional representation/function), compositional inputs/datasets, or compositional representations of inputs. All three are used at various times, which leads to confusion about the scope of what is being studied, especially in earlier parts of the paper before section 3.1 where it becomes clear that $z$ is the disentangled representation which would have been extracted in some earlier part of a model, like an intermediate layer in a neural network. Even after 3.1, though, at times “input” and “representation” are used interchangeably, which causes confusion. This extends even into the discussion section (”dataset statistics and compositional generalization” as opposed to “representation statistics and compositional generalization”).
3. The sections introducing kernel regime and kernel model, in particular in Sections 2 and 3.2, can be cleaned up. Overall, I suggest a restructuring of the content regarding kernel models and your setup. After the introduction, I would suggest immediately introducing your setup and defining variables, as is done currently in Section 3.1, and then describing kernel models within that framework. While doing this, keep a concrete running example that makes it clear what $x$, $z$, $\phi$, and $K$ would refer to on some task that readers would be familiar with. The related work can come after this setup, so that we can make a connection between the related work and your particular setting. Below are some other stray comments about confusions during the introduction of kernel models in your setup.
    1. At times $x$ is used and at other times $z$ is used. Are these the same? Is it a typo in Section 3.2 for instance where you write $f_w(x)$ as opposed to $f_w(z)$?
    2. In equation (1), where do the dual coefficients come from? Are they determined from $w$ and $\phi$ in some way?
    3. Are you making a distinction between “input” and “representation”? Which of these is disentangled in your theory, $z$ or $\phi(z)$? At times, especially earlier on in the paper like in Section 2, it is not obvious.
4. Section 3.1 says that the target $y$ is given by an arbitrary function of $z$ and that your framework is agnostic to this function. This is a bit misleading, as the core contribution of the work is to formalize what downstream functions kernel models can compositionally generalize to w.r.t. some observed disentangled features.
5. Section 4.2 defines conjuction-wise additive functions, but it is quite difficult to quickly parse the math and get intuition for it. Intuition for the proof is given, but not intuition for what a conjunction-wise additive function is. This is a shame because after spending enough time to digest the definition, it is intuitively quite simple. Please try and provide more helpful intuition, as well as an example of a conjunction-wise additive function and how it differs from an additive function.
6. The theoretical and empirical results in Section 5.1 seem to be of significant consequence. Specifically, Proposition 5.1 seems to suggest that very deep neural networks in the kernel regime are unlikely to generalize compositionally as they only represent the full conjunction of disentangled features (if I understand correctly, another way of stating this is that they memorize the training data). While this is explored further in the subsequent subsections, the immediate consequences of Proposition 5.1 are only unpacked in a single sentence following the proposition. I think this result should be emphasized more, and Section 5.1 should do a better job of foreshadowing/leading the results in subsequence subsections.
    1. Additionally, I don’t think it will be completely obvious to everyone why assigning weight to the full conjunction function amounts to memorization. Maybe it can be more clearly articulated (e.g., the full conjunction is a function of combinations of features that can amount to something like a lookup table, without any ability to generalize).
7. Small point: I recommend citing Schug 2024 http://arxiv.org/abs/2312.15001 at the same place as where you cite Wiedemer 2023.

**Questions:**

1. In Section 3.1 first paragraph, the definition of the disentangled representation is a bit unclear.
    1. Does each component have a finite set of possible values it can take on (e.g., multiple values for the “color” component)?
    2. Are the different possible values within a component orthogonal (e.g., vectors for different colours), or are only the vectors across components orthogonal (e.g., colour and shape representations existing in different subspaces)?
    3. Is $C$ constant across samples? In other words, do the individual components (like color and shape) always apply to each possible input?
2. In Section 5 and subsections within, why is the approach of looking at the kernel and representational salience referred to as “modular”?
3. Section 5.2 refers to a set $\mathcal{W}$, but I can’t seem to find where the meaning of this variable was defined earlier on in the paper. Apologies if I’ve missed it, but what is $\mathcal{W}$?
4. See above weaknesses for other questions.

---

> ### Author Response · Authors · 2024-11-23
>
> Thank you very much for your helpful review. We are glad that you liked our paper. Below we respond to your questions and criticisms. We hope that our answers address your concerns.
>
> **Clarity of our setup**
>
> We appreciate your comments on the clarity of our setup and we agree that our presentation of this was needlessly confusing. We have attempted to clean up this presentation. In particular, we now clarify early on that we generally consider models with compositionally structured inputs and immediately introduce our definition for this class of representations (Definition 3.1). While disentangled models are an example of how such a representation could arise, they are no longer a part of our general setup (and we have removed this part from Fig. 1 as well). This is more consistent with the setup as presented e.g. in Jarvis et al. (2023) and Abbe et al. (2023). We also note that we have introduced a small notational change and now refer to the input as $x$ and the underlying components represented by that input as $z=(z_c)_c$. The input $x$ and the underlying components $z$ are connected by the definition of compositionally structured representations.
>
> Overall, we hope that our revised Section 3.1 clarifies the specific theoretical setup we consider. We are thankful to the reviewer for their very helpful comments on this topic and would appreciate hearing whether you think the new presentation is better.
>
> Below we respond to your specific comments on this topic.
>
> > The paper discusses compositional generalization with respect to compositional representations, but equates compositional representations with disentangled representations. I think it’s fair to say that other compositional structures are possible in additional to disentanglement (...). I would make it very clear in the introduction that you are only considering disentangled representations, and that these are a subset of what might be considered compositional representations more generally.
>
> Thank you for highlighting this. We want to emphasize that our theory captures not just disentangled representations but the broader class of compositionally structured representations -- a point that our previous presentation left unclear. By providing the definition of compositional structure early on, we hope to clarify this. We also agree that this does not capture all kinds of compositional representations and now clarify in the introduction that we define a specific class of compositionally structured representations.
>
> > At times it is unclear whether this work is about compositional tasks (tasks where the optimal solution would necessitate some compositional representation/function), compositional inputs/datasets, or compositional representations of inputs. All three are used at various times, which leads to confusion about the scope of what is being studied, especially in earlier parts of the paper before section 3.1 where it becomes clear that is the disentangled representation which would have been extracted in some earlier part of a model, like an intermediate layer in a neural network. Even after 3.1, though, at times “input” and “representation” are used interchangeably, which causes confusion. This extends even into the discussion section (”dataset statistics and compositional generalization” as opposed to “representation statistics and compositional generalization”).
>
> Thank you for this comment. We have gone through the manuscript to try and standardize our terminology. In general, we now refer to the input $x$ as an input and only as a representation insofar as $x$ represents the underlying components. Further, we always refer to $\phi(x)$ as a representation.
>
> > The sections introducing kernel regime and kernel model, in particular in Sections 2 and 3.2, can be cleaned up. Overall, I suggest a restructuring of the content regarding kernel models and your setup. After the introduction, I would suggest immediately introducing your setup and defining variables, as is done currently in Section 3.1, and then describing kernel models within that framework. While doing this, keep a concrete running example that makes it clear what $x$, $z$, $\phi$, and $K$ would refer to on some task that readers would be familiar with. The related work can come after this setup, so that we can make a connection between the related work and your particular setting.
>
> Thank you for this suggestion. We are a bit hesitant to implement it because we're concerned that the related work section will break the flow between model setup and theory. However, we also see that it would make the discussion of kernel models in the related work more immediately relevant. We're hoping that our current changes have already addressed some of your concerns; in particular, we're now introducing the multi-hot input example immediately at the beginning of Section 3 as a guiding example. But we're also open to making further changes to our presentation of the model setup.

---

> > ### Author Response · Authors · 2024-11-23
> >
> > **Other comments**
> >
> > > Section 3.1 says that the target $y$ is given by an arbitrary function of $z$ and that your framework is agnostic to this function. This is a bit misleading, as the core contribution of the work is to formalize what downstream functions kernel models can compositionally generalize to w.r.t. some observed disentangled features.
> >
> > Thanks for highlighting this. To clarify, we mean that we characterize constraints on the model's generalization behavior regardless of how $y$ relates to the input in the training set. We have changed this sentence to clarify this.
> >
> > > Section 4.2 defines conjuction-wise additive functions, but it is quite difficult to quickly parse the math and get intuition for it. Intuition for the proof is given, but not intuition for what a conjunction-wise additive function is. This is a shame because after spending enough time to digest the definition, it is intuitively quite simple. Please try and provide more helpful intuition, as well as an example of a conjunction-wise additive function and how it differs from an additive function.
> >
> > That's a good idea, we've now added such a paragraph (l. 244-252). Due to the page constraints, we unfortunately had to remove the intuition for the proof to make space. However, we think giving immediate intuition for what a conjunction-wise additive function is more important.
> >
> > > The theoretical and empirical results in Section 5.1 seem to be of significant consequence. Specifically, Proposition 5.1 seems to suggest that very deep neural networks in the kernel regime are unlikely to generalize compositionally as they only represent the full conjunction of disentangled features (if I understand correctly, another way of stating this is that they memorize the training data). While this is explored further in the subsequent subsections, the immediate consequences of Proposition 5.1 are only unpacked in a single sentence following the proposition. I think this result should be emphasized more, and Section 5.1 should do a better job of foreshadowing/leading the results in subsequence subsections.
> >
> > We're glad that you agree with us that these important results and appreciate your suggestions for better communicating their importance. We have extended this discussion to better explain what we mean by memorization and to foreshadow the results of the next section (l. 344-348).
> >
> > > Small point: I recommend citing Schug 2024 http://arxiv.org/abs/2312.15001 at the same place as where you cite Wiedemer 2023.
> >
> > We agree and have changed this.
> >
> > > In Section 3.1 first paragraph, the definition of the disentangled representation is a bit unclear.
> >
> > >    1. Does each component have a finite set of possible values it can take on (e.g., multiple values for the “color” component)?
> > >    2. Are the different possible values within a component orthogonal (e.g., vectors for different colours), or are only the vectors across components orthogonal (e.g., colour and shape representations existing in different subspaces)?
> > >    3. Is $C$ constant across samples? In other words, do the individual components (like color and shape) always apply to each possible input?
> >
> > Yes, $C$ is constant and the set of possible components is finite. We have clarified this points. To clarify question 2, the different possible values within a component should be orthogonal or otherwise have equal similarity to each other (i.e. they can be correlated but they should all have equal correlation). Our re-structuring of section 3 has removed this definition anyway as we now only consider the very concrete multi-hot example or general compositionally structured representations.
> >
> > > In Section 5 and subsections within, why is the approach of looking at the kernel and representational salience referred to as “modular”?
> >
> > We meant to convey that our methodology enables general compositions of different salience analyses with different task analyses, but think the word "modular" only confuses our message in this section. We've therefore removed it.
> >
> > > Section 5.2 refers to a set $\mathcal{W}$, but I can’t seem to find where the meaning of this variable was defined earlier on in the paper. Apologies if I’ve missed it, but what is $\mathcal{W}$?
> >
> > Thank you for pointing this out; $\mathcal{W}$ was indeed missing. $\mathcal{W}$ is the set of rows/columns that are in the training set for symbolic addition.

---

> > > ### Comment · Reviewer_CS3X · 2024-11-23
> > >
> > > Thank you for your responses and for taking my feedback into account! I think the changes made to the paper have made it much stronger (especially clarifying my confusion about whether this only pertained to disentangled representations, and giving instead your definition of compositional structure early on). Since all my concerns have been meaningfully addressed in the updated paper and I think the work is very important to the field of compositionality, I have updated my score accordingly. Congratulations on the great paper!

---

> > > > ### Author Response · Authors · 2024-12-03
> > > >
> > > > Thank you very much for increasing your score further! We are glad that you like the paper and really appreciate your helpful review, which has helped us in improving the paper.

---

### Official Review · Reviewer_kn7u · 2024-10-25

**Soundness:** 3
**Presentation:** 3
**Contribution:** 3
**Rating:** 6
**Confidence:** 4

**Summary:**

The paper theoretically analyzes how the model can generalize downstream tasks when the ground-truth disentangled representations are achievable. The authors prove that different models can generalize well on component-wise additivity tasks. Meanwhile, the models are limited in learning tasks involving conjunction-wise additivity, which makes them hard to generalize. Based on the analysis, the authors figured out two important failure modes, i.e., memorization leakage and shortcut bias. The former is because the model learns too many non-atom features from the training set while the latter is caused by the hidden spurious correlation in the training data. Finally, the authors empirically verify their theory using the behavior of some deep neural network structures. Although the paper studies a very important and novel problem about compositional generalization ability, I find it is a struggle for me to understand the paper in depth. However, I do believe the assumption that perfect disentangled representations are achievable harms the applicability of the paper’s results. So I tend to give a negative evaluation.

**Strengths:**

The paper studies an important question that might be overlooked by many related works in compositional generalization, i.e., when perfect disentangled representations are given, what types of downstream tasks are solvable (or non-solvable) in kernel methods? The paper theoretically analyzes this problem and verifies its theory using different experiments.

**Weaknesses:**

### Weakness:

My main concern in this part is about the gap between theory (with many assumptions) and practical systems. Although the paper proposes several experimental results to support the theory, the following concerns forbid me from giving a positive evaluation of this paper:

- The gap between kernel method assumption and deep neural networks. Analyzing the model’s behavior using a simplified model is acceptable, but I expect more results to show that the gap between theory and practical models is negligible. For example, will the deep network on real image input also have a salience score similar to Figure 2? Can we design an ablation study that directly manipulates the weights of overlapping features, e.g. $f_{12}$, and see its influence on the generalization ability? Although the answer might or might not support the claims, knowing the limits of the proposed theory is beneficial.
- The paper assumes that perfect disentangled representations are accessible, which could be easily violated in practical scenarios. Then, how will the results change when the representations are partially disentangled? If the theoretical analysis under this more practical condition is hard, some experimental results under a non-perfect case would be helpful.
- In section 6, the authors replace the one-hot input with real images, which is good. However, the task is still not so practical. Considering the experiments on more common compositional generalization problems, e.g., the dsprite dataset in [1],  can make the claim stronger.
- The paper has potential and the author did a good job of formalizing an important problem. However, the connection between the experimental part and the theory is still not quite clear. I think adding a high-level summary or conceptual diagram that ties together the main theoretical and empirical contributions of the paper would make it easier to understand the whole story of the paper.

[1] Xu, Zhenlin, Marc Niethammer, and Colin A. Raffel. "Compositional generalization in unsupervised compositional representation learning: A study on disentanglement and emergent language." Advances in Neural Information Processing Systems 35 (2022): 25074-25087.

**Questions:**

- In Figure-1b, why [-2]+[1]=1?
- The paper is a bit abstract for me to understand in depth. Especially, I cannot link the provided theory to practical applications, although section 6 indeed provides some results about deep neural networks and common datasets. I think the presentation of the paper is generally good: in the first several sections, we learn how the compositional generalization task and the kernel model studied in this paper are defined. We also learned the salience score is the main metric for tracking different types of features (e.g., how many ground truth features that this learned feature depends on). With this measurement, we know from proposition 5.1 that the later part of the network prefers using those higher-order features, which match our intuitions well. Then, we learn that memorization leak, which could be captured by those higher-order features, hinders the model’s generalization. After that, I began to feel confused: how could we link the results in Figure 4 to the theory provided in the previous part? I think a more detailed analysis of how results in section 6 are correlated to different parts of the theory (e.g., which proposition, which claim, etc.) would make the paper clearer.

---

> ### Author Response · Authors · 2024-11-23
>
> Thank you very much for your helpful review. Below we respond to your questions and criticisms. We hope that our answers address your concerns.
>
> **Practical relevance of compositionally structured representations**
>
> An important concern of the reviewer was the extent to which compositionally structured representations are practically relevant:
>
> > However, I do believe the assumption that perfect disentangled representations are achievable harms the applicability of the paper’s results.
>
> > The paper assumes that perfect disentangled representations are accessible, which could be easily violated in practical scenarios. Then, how will the results change when the representations are partially disentangled? If the theoretical analysis under this more practical condition is hard, some experimental results under a non-perfect case would be helpful.
>
> We agree that this is an important concern --- indeed, in practice, representations are likely not exactly compositionally structured. In our theory, we wanted to investigate the limitations that arise even in the best-case scenario of a perfectly compositionally structured representation. Indeed, we'd like to emphasize that despite the relative simplicity of this case, its generalization behavior has so far been poorly understood.
>
> The non-perfect case may introduce new issues, but we expect that the limitations highlighted in the compositionally structured case will still affect generalization behavior. In particular, our theorem implies that kernel models with compositionally structured representations implement conjunction-wise additive functions. This means that they are unable to perform tasks that cannot be expressed in these terms, such as transitive equivalence. We expect that the constraint to conjunction-wise additive generalization behavior (and the resulting inability to generalize on transitive equivalence) extends to other representations as well. In the revised manuscript, we have added two new analyses that provide evidence for this claim: a novel theoretical result on randomly sampled, Gaussian representations and an empirical analysis of disentangled DSprites representations, analyzing the 1,800 models studied in Locatello et al. (2019). We give an overview of our insights in the general response. In the revised manuscript, we discuss these analyses in detail in Appendices A.5 and C. Additionally, we now discuss the relationship between our theoretical setup and practically relevant representations at the end of Section 4 (l. 286-298). We thank the reviewer for the suggestion to consider this non-perfect case, and in particular for suggesting the DSprites dataset.
>
> Briefly, we find that in both cases, conjunction-wise additivity provides a good description of average-case behavior. As a result, we find that these models are also incapable of systematic generalization on transitive equivalence. Further, whereas our analysis of the influence of representational geometry provides a good characterization of Gaussian representations, these insights break down much more strongly in the case of the DSprites representations.
>
> We hope that our new presentation explains how our theory relates to practically relevant scenarios and would appreciate your feedback.

---

> > ### Author Response · Authors · 2024-11-23
> >
> > **How are our theoretical and empirical results connected?**
> >
> > > The paper has potential and the author did a good job of formalizing an important problem. However, the connection between the experimental part and the theory is still not quite clear. I think adding a high-level summary or conceptual diagram that ties together the main theoretical and empirical contributions of the paper would make it easier to understand the whole story of the paper.
> >
> > > I think a more detailed analysis of how results in section 6 are correlated to different parts of the theory (e.g., which proposition, which claim, etc.) would make the paper clearer.
> >
> > Thank you for this suggestion. In response, we now provide two new diagrams that we hope will clarify these connections. First, we provide a schematic illustration of our insights in Section 4 (see Fig. 6). Second, we provide a new supplementary figure that pulls together the relevant parts of our theoretical and empirical analysis (Fig. 19). Below, we briefly state how our empirical analysis tests our theoretical insights:
> >
> > - In Section 4, we show that compositionally structured representations yield conjunction-wise additive computations. Empirically, we find that this provides a good description of the deep neural networks' generalization behavior (Fig. 9).
> > - Proposition 5.1 indicates that on symbolic addition: a) generalization behavior should underestimate the ground truth by a proportional factor, b) this slope becomes smaller with more conjunctive representations, and c) this slope depends on the training dataset only in terms of its size. We test these insights in Fig. 4b-d.
> > - Our analysis of the influence of training data on generalization on context dependence suggests that for the smallest training set (CD-3), networks are likely to fail generalization due to a statistical shortcut, whereas this is much less likely on larger training sets (CD-2 and CD-1). We confirm this prediction empirically in Fig. 4e.
> >
> > > The gap between kernel method assumption and deep neural networks. Analyzing the model’s behavior using a simplified model is acceptable, but I expect more results to show that the gap between theory and practical models is negligible.
> >
> > We agree that it is important to discuss the relationship between our theoretical insight and the practical experiments related to it. We emphasize that we are not able to provide exact quantitative bounds on the behavior of deep networks in the feature learning regime and agree that testing the extent to which we are able to describe their behavior is important. In response to your questions, we provided two new analyses.
> >
> > First, we analyzed the relationship between the salience of the convolutional neural networks' neural tangent kernel and their generalization behavior on symbolic addition. We found that while our theory accurately described the qualitative relationship, it did not provide an exact quantitative fit (Fig. 13b). We found that this was because the neural tangent kernel changed over the course of training, indicating that the networks are trained in the feature learning regime (Fig. 13a). We appreciate your suggestion as it allows us to concretize the scope of our theory with respect to these empirical neural networks. Specifically, we aim to describe a range of qualitative phenomena (as listed in the response above), but leave a more exact quantitative characterization to future work. We have added a sentence to the discussion to clarify that (l. 531-533).
> >
> > Second, our empirical experiments so far have tested whether our theory provides useful insights into the relationship between dataset statistics and compositional generalization. To test whether it also captures the effects of certain architectural interventions (as per your suggestion), we varied the number of fully connected layers in the convolutional neural network trained on symbolic addition. Our theory would predict that deeper networks have more conjunctive representations (see Fig. 2 and Proposition 5.1) and therefore worse compositional generalization behavior on symbolic addition. Consistent with this prediction, we indeed found that deeper networks (despite generalizing slightly better in-distribution) had impaired compositional generalization. Thank you for this suggestion!
> >
> > **Minor comments**
> >
> > > In Figure-1b, why [-2]+[1]=1?
> >
> > That was a typo, it should say [-2]+[1]=-1. Thank you for pointing this out!
> >
> > > I think the presentation of the paper is generally good (...)
> >
> > Thank you for your positive assessment of the presentation in these parts of the paper. It's useful for us to know in what parts of our paper our explanations were clear; we also appreciate your detailed explanations of what you found less clear and hope that our changes can address these concerns.

---

> > > ### Comment · Reviewer_kn7u · 2024-11-24
> > >
> > > Thanks very much for the author's response, which addresses most of my concerns well. The added experiments make the paper stronger. I like the discussions and more demonstration figures in the Appendix part, which makes the paper and experiments easier to understand. Although the analysis in the non-perfect disentanglement representation case is important in this direction, I agree with the authors that figuring out the generalization capability limitations under the perfect setting is a very important starting point. I hence raised my score from 5 to 6 and my confidence from 3 to 4. I'm looking forward to seeing the new version of the paper.

---

> > > > ### Author Response · Authors · 2024-12-03
> > > >
> > > > Thank you very much for your positive evaluation of our response. We are glad that we were able to address most of your concerns and really appreciate your helpful review, which has helped us improve our paper.

---

### Official Review · Reviewer_Gayd · 2024-11-02

**Soundness:** 4
**Presentation:** 3
**Contribution:** 3
**Rating:** 8
**Confidence:** 3

**Summary:**

This paper considers the question, "Given a compositional (i.e., disentangled) representation, under what circumstances can kernel models (e.g., a linear readout trained with SGD) generalize compositionally?" The usefulness of disentangled representation for compositional generalization has been debated in the literature, but generally from an empirical angle or specific settings (e.g., object-centric learning). The authors opt to study the problem theoretically from the perspective of kernel models. This approach leads to a reasonably general framework that highlights important limitations: The type of computation that can be generalized is restricted to simple additive combinations of components, what the authors call "conjunction-wise additive". Even within this class of tasks, the authors go on to show that a kernel model will often not generalize perfectly, as it is biased to consider (spurious) interactions between all components or fall for spurious shortcuts.

**Strengths:**

This paper presents a comprehensive study of compositional generalization. While the setting is restricted to fixed, disentangled representations (which, in practice, are not guaranteed to be recovered by the model), it succeeds in abstracting compositional generalization to encompass many specific problem formulations. The focus on kernel methods allows the authors to draw insightful conclusions about the limits of compositional generalization in terms of the operations that can be learned, which are entirely novel to the best of my knowledge. It is nice to see that the biases predicted by the theory can be demonstrated on real (if toyish) datasets.

**Weaknesses:**

The paper is quite dense and not easily accessible to readers. I can see that the authors attempted to illustrate their theoretical findings on a few example tasks to convey some intuition, but the explanation of the example tasks is still frequently very technical and hard to parse. For example, it is still somewhat unclear to me why transitive equivalence is ruled out by Thrm. 4.1 (see also minor suggestions below).

Given Thrm. 4.1, I would have appreciated a short discussion on existing compositional generalization tasks from the literature, e.g., from Schott et al., or Locatello et al., or Hupkes et al. Which of these tasks would be solvable by a kernel model? Could the insights from this paper aid in understanding why the existing literature on the usefulness of disentangled representations is so divided?

**Questions:**

# Questions
- I'm not sure Wiedemer et al. (2023b) is characterized correctly as requiring a network that is constrained to be linear/additive. If I recall correctly, this paper allowed for arbitrary combinations of components. If so, a more detailed comparison of the assumptions in that work would be helpful to assess the new insights from this work.
- LL269: Why does $f_{12}(z)$ "fall away"? I assume because $f_J(z_{12}) = 0$, but why? Could you elucidate this on a brief example?
- §5.1 / §B: Even after reading these sections, it is unclear to me how the representational salience is computed in practice. Could you walk me through an example with a batch of training points?
- §5.1 / Fig. 2: Do you have any intuition where the difference between nonlinearities is coming from?
- §6: Am I correct in assuming all models are randomly initialized and not trained? This is not entirely clear from the text. Also, what specific model architectures were used, and how are they initialized? What are the training settings for the linear readout? As the code will not be published, these details are essential to ensure reproducibility. I recommend adding a corresponding section detailing initialization, architecture, and other specifications to the main text or appendix.

# Minor suggestions
- LL99: Why call a combination of components a "conjunction", and not the more obvious "composition"?
- Fig. 1c: It was not immediately clear which side of the training set line is the training/test set
- LL256: "The readout weight ..." is unclear, is there a word missing? When will it change?
- L269: should be "test set"
- LL317: reusing $c$ here as a number of components is slightly confusing when $c$ was used above as a component index. I recommend using $k$ or $n$ instead
- LL354: What is $\mathcal W$ here? It has not been introduced before

---

> ### Author Response · Authors · 2024-11-23
>
> Thank you very much for your helpful review. Below we respond to your questions and criticisms. We hope that our answers address your concerns.
>
> > §6: Am I correct in assuming all models are randomly initialized and not trained? (...) As the code will not be published, these details are essential to ensure reproducibility. I recommend adding a corresponding section detailing initialization, architecture, and other specifications to the main text or appendix.
>
> Importantly, these models are trained with backpropagation and all weights in those models are trained, in the case of the ConvNet and the ResNet using SGD and in the case of the ViT using Adam. They are all initialized using He initialization. We provide further details on their architecture in Appendix C.1 and have made this description more extensive in the revised manuscript. Additionally, we will also upload our codebase to ensure reproducibility. Thank you for pointing this out and we apologize for the confusion.
>
> > The paper is quite dense and not easily accessible to readers. I can see that the authors attempted to illustrate their theoretical findings on a few example tasks to convey some intuition, but the explanation of the example tasks is still frequently very technical and hard to parse. For example, it is still somewhat unclear to me why transitive equivalence is ruled out by Thrm. 4.1 (see also minor suggestions below).
>
> > LL269: Why does $f_{12}(z)$ "fall away"? I assume because $f_J(z_{12})=0$, but why? Could you elucidate this on a brief example?
>
> Thank you for pointing this out that this could be presented in more intuitive terms. To make the connection to the theorem explicit, this is because for test set items, the conjunction has never been seen during training and so is not a part of the sum in Eq. 4. As a result the model cannot use the conjunctive term $f_{12}(z)$. More intuitively, the unseen conjunctive terms correspond to a direction in representational space that was not seen during training (as this specific combination of components was never seen) and, as a result, the model did not change its weights in this direction and they remained at their initial value of zero. We now provide a novel diagram in the appendix (Fig. 5) that aims to convey the central insights of Section 4. We have also added a paragraph directly after the theorem explaining its central insights (l. 244-252).
>
> > Given Thrm. 4.1, I would have appreciated a short discussion on existing compositional generalization tasks from the literature, e.g., from Schott et al., or Locatello et al., or Hupkes et al. Which of these tasks would be solvable by a kernel model? Could the insights from this paper aid in understanding why the existing literature on the usefulness of disentangled representations is so divided?
>
> We agree that it would be useful to further link Theorem 4.1 to the literature. Both Locatello et al. and Schott et al. consider as their downstream task direct decoding of the underlying components. This is a conjunction-wise additive (and indeed component-wise additive) computation. However, linear readout models may still fail to generalize on these tasks due to a memorization leak and a shortcut bias. Indeed, we analyze two simple instances of this problem in Appendix D.3 (invariance and partial exposure) and find that they are affected by these failure modes. Intriguingly, Schott et al. also report a regression to the mean (Section 5.3 in their paper) which is consistent with memorization leaks and shortcut biases. Our paper theoretically explains why this regression to the mean may arise in the linear readout setting. We have added a sentence explaining this connection (l. 396-399).
>
> Our work is related to existing literature on disentangled representations in that we find that subtle differences in representational geometry and training data (in particular on the context dependence task) can yield substantially different generalization behavior: for example, the darkgreen region in representational geometry space in Fig. 3c robustly generalizes on CD-3, whereas the lightgreen region consistently does not generalize. Depending on the exact nonlinearity and depth chosen, neural network representations may either be in one of these regions or the other. This emphasizes how subtle changes in experimental settings can substantially change conclusions --- as disentangled representations are often evaluated in terms of a linear readout, this may explain why empirical evaluations of these representations have often been inconsistent between different experimental setups. We briefly mention this in l. 414-419 and have added a sentence making this connection explicit. Thank you for pointing out these connections. We are quite excited about the ways in which our theory may speak to these existing lines of research and appreciate the opportunity to clarify these connections.

---

> > ### Author Response · Authors · 2024-11-23
> >
> > > I'm not sure Wiedemer et al. (2023b) is characterized correctly as requiring a network that is constrained to be linear/additive. If I recall correctly, this paper allowed for arbitrary combinations of components. If so, a more detailed comparison of the assumptions in that work would be helpful to assess the new insights from this work.
> >
> > Thank you for pointing this out. We agree and have modified our phrasing. To clarify, Wiedemer et al. consider a known composition function, which imposes constraints on the set of functions their model can learn. In contrast, our models can learn arbitrary training data and we investigate the constraints on test set generalization that emerge in spite of this lack of constraints.
> >
> > > §5.1 / §B: Even after reading these sections, it is unclear to me how the representational salience is computed in practice. Could you walk me through an example with a batch of training points?
> >
> > Thank you for highlighting this. We have added such a walkthrough to Appendix B.1. Briefly, we can write the trial-by-trial similarity between two inputs as a function of the set of overlapping components between those inputs (as all pairs of inputs with the same overlapping components have equal similarity in a compositionally structured input). We then compute the salience by recursively subtracting the salience of all subsets of each of those sets of components (this is defined in Eq. (14). In particular, the salience of the empty set is simply the similarity between two inputs with no overlapping components: $\overline{S}(\emptyset)=\text{Sim}(\emptyset)$. The salience of a single component, $\{c\}$, is given by the similarity between two inputs overlapping in this component, subtracting the salience of the empty set: $\overline{S}(\{c\})=\text{Sim}(\{c\})-\overline{S}(\emptyset)$. The salience of a conjunction of two components is given by $\overline{S}(\{1,2\})=\text{Sim}(\{1,2\})-(\overline{S}(\{1\})+\overline{S}(\{2\})+\overline{S}(\emptyset))$ and so on. Finally, we normalize the salience so that the saliences of all non-empty conjunctions sum to one. We hope that this clarified the concrete steps; please let us know if there are any remaining confusions about this.
> >
> > > §5.1 / Fig. 2: Do you have any intuition where the difference between nonlinearities is coming from?
> >
> > Intuitively, different nonlinearities impact two inputs with a given similarity differently. For example, the rectified quadratic nonlinearity strongly amplifies the similarity between more similar inputs compared to less similar inputs, whereas the ReLU nonlinearity does this less. As a result, the rectified quadratic nonlinearity further amplifies the similarity between inputs overlapping in more than one component, whereas it takes repeated applications of the ReLU function to have the same effect.
> >
> > > LL99: Why call a combination of components a "conjunction", and not the more obvious "composition"?
> >
> > We used the word conjunction as it is used for this exact concept in cognitive science and neuroscience. Our concern with the word "composition" was that this word has many meanings in the context of our study.
> >
> > > L269: should be "test set"
> >
> > To clarify, in this paragraph, we first consider what the effect of the conjunction-wise additive function specified in Eq. (3) is on the training set, to illustrate that there are no constraints, before moving on to the test set in the next line.
> >
> > > LL354: What is $\mathcal{W}$ here? It has not been introduced before
> >
> > Thank you for pointing this out; $\mathcal{W}$ was missing in l. 156. We've now corrected this. To clarify, $\mathcal{W}$ is the set of rows/columns that are in the training set for symbolic addition.

---

### Official Review · Reviewer_jpBR · 2024-11-06

**Soundness:** 2
**Presentation:** 2
**Contribution:** 2
**Rating:** 3
**Confidence:** 3

**Summary:**

This paper studies compositional generalization from the perspective of kernel theory, showing that they are constrained to adding up values assigned to each combination seen during training. This result demonstrates a fundamental restriction in the generalisation capabilities of kernel models. The theory is validated empirically and shows, that the theory also captures certain behaviors of deep neural networks trained on compositional tasks.

**Strengths:**

The work tries to establish a stronger theory of compositional generalization in kernel and neural network models, addressing a crucial white spot in our understanding of statistical machine learning models. It also takes on an original angle to the problem by studying compositionality in the kernel limit, and I appreciate the empirical evaluation and comparison w.r.t. to deep neural networks.

**Weaknesses:**

- There is a wide gap between the claims of the paper and what is actually shown and proved. In particular, the theory only considers kernel models with random weights networks as feature extractors. In this limit the intermediate representation is compositionally structured (assuming the input is as well). But that doesn't necessarily extend to other kernels as implied by the abstract, which doesn't mention this limitation once.

- Also, while the work tries hard to make a connection to disentangled networks and thus establish an empirical relevance (which I generally welcome), the paper very carefully states that "This highlights fundamental computational restrictions on [...] pretrained models with disentangled representations that are fine-tuned in the kernel regime and infinite-width neural networks". For one, that's overly broad because even if a DNN is trained on disentangling the representation, there is no guarantee it is disentangled outside of the training data (in contrast to what is considered in the theoretical results). Second, this only applies if also the input is already compositional, rendering the statement mostly irrelevant for any practical application (e.g., in vision or language you don't have a compositional input to start with).

- The overall presentation is quite confusing and hard to follow and I am having a hard time even understanding what the precise contribution of the paper is, what assumptions are being made and what conclusions can be drawn from it. In particular, what is drawn from the results is mixed up with the precise theoretical results. It would be great to have one section that is really precise and doesn't immediately tries to generalise to pre-trained models, before outlining precisely how and in what way the results might generalise to empirical models and tasks.

- There are pretty much no details on the empirical experiments with deep neural networks (both in the main text and the appendix). That part is thus almost impossible to evaluate. However, if my assumptions are correct, then the DNNs are trained in extremely simplistic settings that resemble nothing even close to what the networks are actually used for in practice (despite the use of MNIST or CIFAR images). Hence, claiming that the theory can explain the empirical behavior of DNNs is misleading.

**Questions:**

- Line 262 states that "kernel models cannot solve any task that cannot be expressed in a conjunction-wise additive terms" - can you explain which tasks cannot be phrased in this way? In the worst case, the full conjunction can express any non-linear relationship, no?

- Line 269: why would the term f_{12}(z) fall away on the test set? I don't see how this should happen.

- How can a compositionally structured kernel model learn arbitrary training data, as is implied by section 4.3?

- Please add all the details on your empirical evaluation.

- In the related work, you state that Wiedemer et al. and Lachapelle et al. is restricted to linear networks, but you probably mean linear interactions between the components.

- Regarding the conjunction-wise additive computation: Why is the result surprising? In line 464 it states that "neural networks tend to implement conjunction-wise additive computations - at least when trained on conjunction-wise additive tasks". It would be quite surprising if it wouldn't implement that, no?

My scores currently reflect that the contributions of the paper are very difficult to assess and that the claims are way too broad (at least that's my current understanding). I can see interesting aspects and would like to encourage the authors to outline and state their contributions very clearly and without overclaiming. Understanding generalisation in machine learning is hard, so even some understanding on random-weights models is definitely interesting.

---

> ### Author Response · Authors · 2024-11-23
>
> Thank you very much for your helpful review. We are glad that you agree that understanding compositional generalization is an important problem and appreciate your constructive feedback. We hope that the response below addresses your questions and your criticisms. We'd love to continue discussing these points; in particular, we'd appreciate your thoughts on our clarified theoretical contributions (directly below).
>
> **Clarifying our contributions**
>
> > The overall presentation is quite confusing and hard to follow and I am having a hard time even understanding what the precise contribution of the paper is, what assumptions are being made and what conclusions can be drawn from it. In particular, what is drawn from the results is mixed up with the precise theoretical results. It would be great to have one section that is really precise and doesn't immediately tries to generalise to pre-trained models, before outlining precisely how and in what way the results might generalise to empirical models and tasks.
>
> Thank you for this comment. In light of it, we wanted to briefly state our main theoretical contribution here as clearly as possible, and also make explicit our assumptions and their limitations. We have also modified the text and introduced a schematic figure (Fig. 5) to the manuscript in order to make presentation more clear.
>
> - We introduce **compositionally structured representations**, which are defined as representations whose trial-by-trial similarity only depends on the number of overlapping components between the two trials (now defined in Definition 3.1).
> - We then analyze the compositional generalization of kernel models instantiated by first transforming these representations by deep neural networks, then learning the weights on a linear readout of this representation. This allows us to capture the representational changes that happen as a compositional representation is modified by a neural network, what kind of compositional generalization is possible following learning on the resulting representation.
> - We find that the representations remain compositionally structured after having been processed by the deep neural network (in the infinite-width limit) (now stated in Proposition 4.1).
> - We find that kernel models with compositionally structured representations are limited to a **conjunction-wise additive computation** on the test set (Theorem 4.2). This refers to any compositional computation that can be performed by summing compositional features and combinations of features seen during training, and can capture a suprisingly large number of compositional behaviors. This allows us to identify which tasks are fundamentally unsolvable by these models and thus require different learning mechanisms. We spell out these consequences in Section 4.3.
> - In Section 5, we then analyze in more detail how the training data and the deep neural network influence the generalization behavior of these models on conjunction-wise additive tasks. Although these tasks can be performed by a conjunction-wise additive computation, we identify two failure modes that can still disrupt generalization by preventing kernel models from discovering the right computation: memorization leak and shortcut bias.
> - These failure modes result from imbalances in the training data, and we note that deep neural networks show the same qualitative sensitivity to training data imbalances as well.
>
> We appreciate your suggestions on how to make our contributions in Section 4 clearer and have modified our manuscript accordingly. In particular, we now introduce compositionally structured representations when first introducing the task setup (i.e. in Section 3.1) and further present all of our theoretical contributions in theorem environments (whereas Definition 3.1 and Proposition 4.1 were previously just stated in paragraphs). We have also removed discussion of potential applications of our theory from this section in order to focus on explaining our concrete results. Instead, we have now added a final paragraph to the section where we discuss the relationship between our theory and practically relevant scenarios, and in particular clarify when our theory cannot directly speak to these scenarios. Thank you for this suggestion.

---

> > ### Author Response · Authors · 2024-11-23
> >
> > > Line 262 states that "kernel models cannot solve any task that cannot be expressed in a conjunction-wise additive terms" - can you explain which tasks cannot be phrased in this way? In the worst case, the full conjunction can express any non-linear relationship, no?
> >
> > For inputs with two components, the model is constrained to adding up a value for each component and cannot encode a nonlinear interaction between components. The model cannot use the full conjunction on the test set, as it has not seen this full conjunction (for test set inputs) during training. In particular, this model can therefore not generalize on the transitive equivalence task. A similar limitation arises for tasks with more than two components. Note that conjunction-wise additivity refers specifically to the computation defined in Eq. 4 in Theorem 4.2. In the revised manuscript, we discuss these implications immediately after introducing our theorem (l. 244-252).
> >
> > > Line 269: why would the term f_{12}(z) fall away on the test set? I don't see how this should happen.
> >
> > To make the connection to the theorem explicit, this is because for test set items, the conjunction has never been seen during training and so is not a part of the sum in Eq. 4. As a result the model is constrained to summing up $f_{1}(z_1)+f_2(z_2)$. More intuitively, the unseen conjunctive terms correspond to a direction in representational space that was unseen during training (as this specific combination of components was never seen). As a result, the model did not change its weights in response to this direction and they remained at their initial value of zero.
> >
> > **Clarifying the scope of our theory**
> >
> > > There is a wide gap between the claims of the paper and what is actually shown and proved.
> >
> > Thank you for these comments. We have modified the abstract to clarify that we are talking about kernel models with compositionally structured representations and to clarify our assumptions and their limits. We have also added a sentence on this to the discussion. In particular, our theory can only speak to compositionally structured representations. In practice, representations will often not be exactly compositionally structured, for example due to general noisiness in the representations or because certain components are more similar to each other (e.g. perhaps certain shapes are more similar to each other than to other shapes). For some of these settings, we empirically observe that models are somewhat well described by our theory; for others we expect this may not be true. We have added a paragraph discussing these relationships to Section 4 (l. 286-298); we further note that we now provide two new analyses on non-compositionally structured representations (see the overview in the general response). We hope that these analyses help us more explicitly tie our insights to a broader range of representations, while also making very clear in what settings we can make rigorous claims. We hope our new presentation of our assumptions is more clear and avoids overclaiming, and would appreciate your feedback.
> >
> > > There are pretty much no details on the empirical experiments with deep neural networks (...) However, if my assumptions are correct, then the DNNs are trained in extremely simplistic settings (...) Hence, claiming that the theory can explain the empirical behavior of DNNs is misleading.
> >
> > Thank you for pointing this out. We now describe the setup in more detail (see Appendix D). If you think any of this would be important to mention in the main text, we'd happily adjust the draft accordingly. Briefly, we consider each image category as corresponding to one component and assume an input that concatenates each component's image. For example, on one task instance of symbolic addition, images with the handwritten digit "0" may correspond to the magnitude -2 and images with the handwritten digit "4" may correspond to the magnitude 3 (the role of each digit category is randomly sampled for each task instance). The target output associated with this input (see the picture in Fig. 4a) then consists in the sum of these magnitudes (i.e. 1). We can therefore use this setup to investigate compositional generalization to novel combinations of components (i.e. image categories). All networks are relevant large-scale neural network architectures (ConvNets, ResNets, and Transformers) that are trained with backpropagation.
> >
> > To emphasize that we are not making any exhaustive claims about neural network behavior, we have changed the section heading to "Our theory can describe the behavior of deep networks on conjunction-wise additive tasks." While we agree that our tasks are somewhat artificial, we'd like to note that this is a fairly large-scale setup for a paper whose primary focus is theoretical, and indeed similar to previous experiments in theoretical papers (see e.g. Jarvis et al., 2023). We'd appreciate any further pointers as to what about this setting you consider overly simplistic.

---

> > > ### Author Response · Authors · 2024-11-23
> > >
> > > **Further responses**
> > >
> > > > Also, while the work tries hard to make a connection to disentangled networks and thus establish an empirical relevance (which I generally welcome), the paper very carefully states that "This highlights fundamental computational restrictions on [...] pretrained models with disentangled representations that are fine-tuned in the kernel regime and infinite-width neural networks". For one, that's overly broad because even if a DNN is trained on disentangling the representation, there is no guarantee it is disentangled outside of the training data (in contrast to what is considered in the theoretical results). Second, this only applies if also the input is already compositional, rendering the statement mostly irrelevant for any practical application (e.g., in vision or language you don't have a compositional input to start with).
> > >
> > > Thank you for this comment. As we note above, we have added further discussion to indicate that the limitations on compositional generalization arising in compositionally structured representations may affect a broader range of linear readout models (or models trained in the kernel regime) as well. Our results therefore indicate that even in what may be considered the ideal-case scenario (a model that has learned perfectly disentangled representations), the compositional generalization behavior of models using this representations has some fundamental limitations.
> > >
> > > > How can a compositionally structured kernel model learn arbitrary training data, as is implied by section 4.3?
> > >
> > > It can do so by using the full conjunction, which is specific to each training data point. Technically, this depends on the salience of that full conjunction being nonzero; as we explain in Section 4.1, an additive input representation cannot learn arbitrary training data. However, any of the neural network transformations we consider yields a nonzero salience for the full conjunction; thus, as long as the model has such a nonlinear transformation, it can learn arbitrary training data. This is related to various universal function approximation theorems, as we note in l. 219-221.
> > >
> > > > In the related work, you state that Wiedemer et al. and Lachapelle et al. is restricted to linear networks, but you probably mean linear interactions between the components.
> > >
> > > Thanks, we've changed the phrasing.

---

### Author Response · Authors · 2024-11-23

We'd like to thank all the reviewers for their very helpful reviews. Below we respond to each of them individually. Here we provide a brief overview of the large-scale changes that we've made to the manuscript (changes are highlighted in blue; to the extent that e.g. line numbers have changed, we will refer to the line numbers in the revised manuscript).

**Clarifying how our theory relates to practically relevant representations**

Reviewer kn7u noted their concerns that "the assumption that perfect disentangled representations are achievable harms the applicability of the paper’s results". On the other hand, reviewer jpBR noted that while "even some understanding on random-weights models is definitely interesting", they wanted us to more clearly state our contributions without overclaiming. To address both of these concerns, we a) updated our manuscript to more clearly clarify the scope of our contributions and the fact that our theory is limited to compositionally structured representation (see our response to reviewer jpBR).

Further, we b) added two new analyses (one theoretical and one empirical) to clarify how our theoretical insights may apply to non-compostionally structured representations. Specifically, we first considered randomly sampled Gaussian representations that are only compositionally structured in expectation. In a minor extension of our theory, we prove that models trained on such representations are, in expectation, also conjunction-wise additive (Proposition A.2). This demonstrates that our theory is robust to random deviations from compositional structure. We empirically confirmed this proposition in simulations (Fig. 6a). In particular, this implies that these representations can also not systematically generalize on transitive equivalence, which we confirmed empirically (Fig. 6b). We then investigated whether our insights on how representational geometry influences compositional generalization generalize to this novel setting. In simulations, we found that our theory indeed captures the behavior of models averaged across many sampled representations (Fig. 6c,d).

Second, we considered the DSprites dataset, analyzing the representations emerging in 1,800 disentangled representation learning models (as analyzed by Locatello et al., 2019). For each of these models, we considered fifty randomly sampled instances of each task, where we sampled the role played by the different components. In simulations, we found that the average model across these task instances was generally well described by a conjunction-wise additive computation (Fig. 8a) and indeed, these representations were also uniformly unable to systematically generalize on transitive equivalence (Fig. 8b). In contrast to the randomly sampled representations, however, we found that our theory did not capture the more fine-grained generalization behavior of these models on symbolic addition (Fig. 8c).

Overall, these new analyses contextualize our theory of compositional generalization in compositionally structured representations by a) demonstrating that conjunction-wise additivity is a compositional generalization class that is relevant beyond compositionally structured representations. The DSprites dataset also b) illustrates, however, that the theory we present in Section 5 does not directly generalize to the non-compositionally structured case. We are grateful to the reviewers for articulating their concerns about this topic and hope that these results can address them. We discuss our analyses in detail in Appendices A.5 and C and summarize the most important insights in the main text in a paragraph at the end of Section 4 (l. 286-298).

---

> ### Author Response · Authors · 2024-11-23
>
> **Improving presentation of the theoretical setup**
>
> The reviewers pointed out several aspects of the presentation they thought could be improved. We respond to their suggestions in detail in the individual responses and give an overview here. The biggest change we've made to our presentation is in Section 3.1. Specifically, we now immediately define our class of compositionally structured representations (Definition 3.1; previously defined in Section 4), rather than discussing only disentangled input representations. This enables us to clarify the scope of our theory from the beginning. As part of these changes, we now also explicitly separate the components underlying the input (which we continue to denote by $z=(z_c)_c$) and the input itself (which we now denote by $x\in\mathbb{R}^d$). To further clarify our theoretical insights in Section 4, we have also added a new schematic figure illustrating these insights (Fig. 5). Finally, we now also provide a more concrete intuition about what a conjunction-wise additive computation is (l. 244-252). We note that to accommodate for these changes, we moved section 7 (the brief outlook on the role of feature learning) fully into the appendix (Appendix F).
>
> Overall, we really appreciate everyone's detailed comments on which aspects they found clearly explained and which aspects they'd like to see improved. We hope that our revised manuscript addresses the reviewers' concerns and would appreciate their further feedback.

---

### Meta-Review · Area_Chair_mFbU · 2024-12-18

**Metareview:**

(a) Summary

This paper investigates compositional generalization(CG) in kernel models and deep neural networks with disentangled input. Theoretically, it presents the constraints for kernel models to solve a range of compositional tasks (conjunction-wise additive tasks). Empirically, it demonstrates that the theory captures the CG behavior of DNN models on the proposed compositional tasks.


(b) Strengths
+ This paper studies CG from theoretical and kernel angle. It succeeds in abstracting compositional generalization to encompass many specific problem formulations.
+ It conducts empirical evaluation and comparison of DNNs to support the theoretical claims.
+ It draws insightful conclusions about the limits of compositional generalization in terms of the operations that can be learned.
+ The biases predicted by the theory can be demonstrated on real  datasets.

(c) Weaknesses
- The assumption that perfect disentangled representations are achievable harms the applicability of the paper’s results.
- There is a gap between the simple models the theory describes and actual DNNs.
- The paper is too dense to read and the presentation could be improved.
are missing.

(d) decision

The paper presents theoretically-sound contributions to the field of compositional representations and generalization. While there is a gap between the simple models the theory describes and actual DNNs, this is to be expected given the complexity of real DNNs. Considering it's worth to share the theory paper in the community to inspire discussion, I recommend accept.

**Additional Comments On Reviewer Discussion:**

The reviewers acknowledged good motivation and theoretical analysis of CG. There are shared concerns on the clarity of the paper and missing details on the empirical experiments. The authors' rebuttal and revised manuscript resolved the above concerns. Three reviewers considered the authors's rebuttal resolved their concerns and raised their scores, while Reviewer jpBP didn't respond to the comments.

---

### Decision · Program_Chairs · 2025-01-22

Accept (Poster)